# *Cis*-activation in the Notch signaling pathway

Nagarajan Nandagopal[†], Leah A Santat[†], Michael B Elowitz*

Division of Biology and Biological Engineering, California Institute of Technology, Howard Hughes Medical Institute, Pasadena, United States

**Abstract** The Notch signaling pathway consists of transmembrane ligands and receptors that can interact both within the same cell (*cis*) and across cell boundaries (*trans*). Previous work has shown that *cis*-interactions act to inhibit productive signaling. Here, by analyzing Notch activation in single cells while controlling cell density and ligand expression level, we show that *cis*-ligands can also activate Notch receptors. This *cis*-activation process resembles *trans*-activation in its ligand level dependence, susceptibility to *cis*-inhibition, and sensitivity to Fringe modification. *Cis*-activation occurred for multiple ligand-receptor pairs, in diverse cell types, and affected survival in neural stem cells. Finally, mathematical modeling shows how *cis*-activation could potentially expand the capabilities of Notch signaling, for example enabling 'negative' (repressive) signaling. These results establish *cis*-activation as an additional mode of signaling in the Notch pathway, and should contribute to a more complete understanding of how Notch signaling functions in developmental, physiological, and biomedical contexts.
DOI: https://doi.org/10.7554/eLife.37880.001

*For correspondence:
melowitz@caltech.edu

[†]These authors contributed equally to this work

**Competing interests:** The authors declare that no competing interests exist.

## Introduction

The Notch signaling pathway enables intercellular communication in animals. It plays critical roles in diverse developmental and physiological processes, and is often mis-regulated in disease, including cancer (*Louvi and Artavanis-Tsakonas, 2012*; *Siebel and Lendahl, 2017*). Notch signaling occurs when membrane-bound ligands such as Dll1 and Dll4 on one cell activate Notch receptors on neighboring cells (*Figure 1A*, *trans*-activation) (*Artavanis-Tsakonas et al., 1999*; *Nichols et al., 2007b*; *Bray, 2016*). However, other types of interactions are also known to occur. Intercellular interactions between Notch1 and the ligand Jag1 have been shown to block *trans*-activation during angiogenesis and in cell culture (*Figure 1A*, *trans*-inhibition) (*Benedito et al., 2009*; *Hicks et al., 2000*; *Golson et al., 2009*). Additionally, Notch ligands and receptors co-expressed in the same cell have been shown to mutually inhibit one another, suppressing productive intercellular signaling (*Figure 1A*, *cis*-inhibition) (*Sprinzak et al., 2010*; *del Álamo et al., 2011*; *Fiuza et al., 2010*). Such '*cis*-inhibition' has been shown to be important in diverse developmental processes including neurogenesis, wing margin formation in Drosophila, and maintenance of postnatal human epidermal stem cells (*Micchelli et al., 1997*; *Jacobsen et al., 1998*; *Franklin et al., 1999*; *Lowell et al., 2000*).

The ability of co-expressed Notch ligands and receptors to interact on the same cell provokes the question of whether such interactions might also lead to pathway activation (*Figure 1A*, '*cis*-activation'). *Cis*-activation has been postulated (*Formosa-Jordan and Ibañes, 2014a*; *Hsieh and Lo, 2012*; *Coumailleau et al., 2009*; *Pelullo et al., 2014*), but has not been systematically investigated. A key challenge in identifying and characterizing such a behavior is the difficulty of discriminating between *trans*- and *cis*-activation in a multicellular tissue context; that is, attributing any observed Notch signal to *trans* or *cis* ligand-receptor interactions. It has therefore remained unclear whether and where *cis*-activation occurs, how it compares to *trans*-activation, and how it might co-exist with *cis*-inhibition.

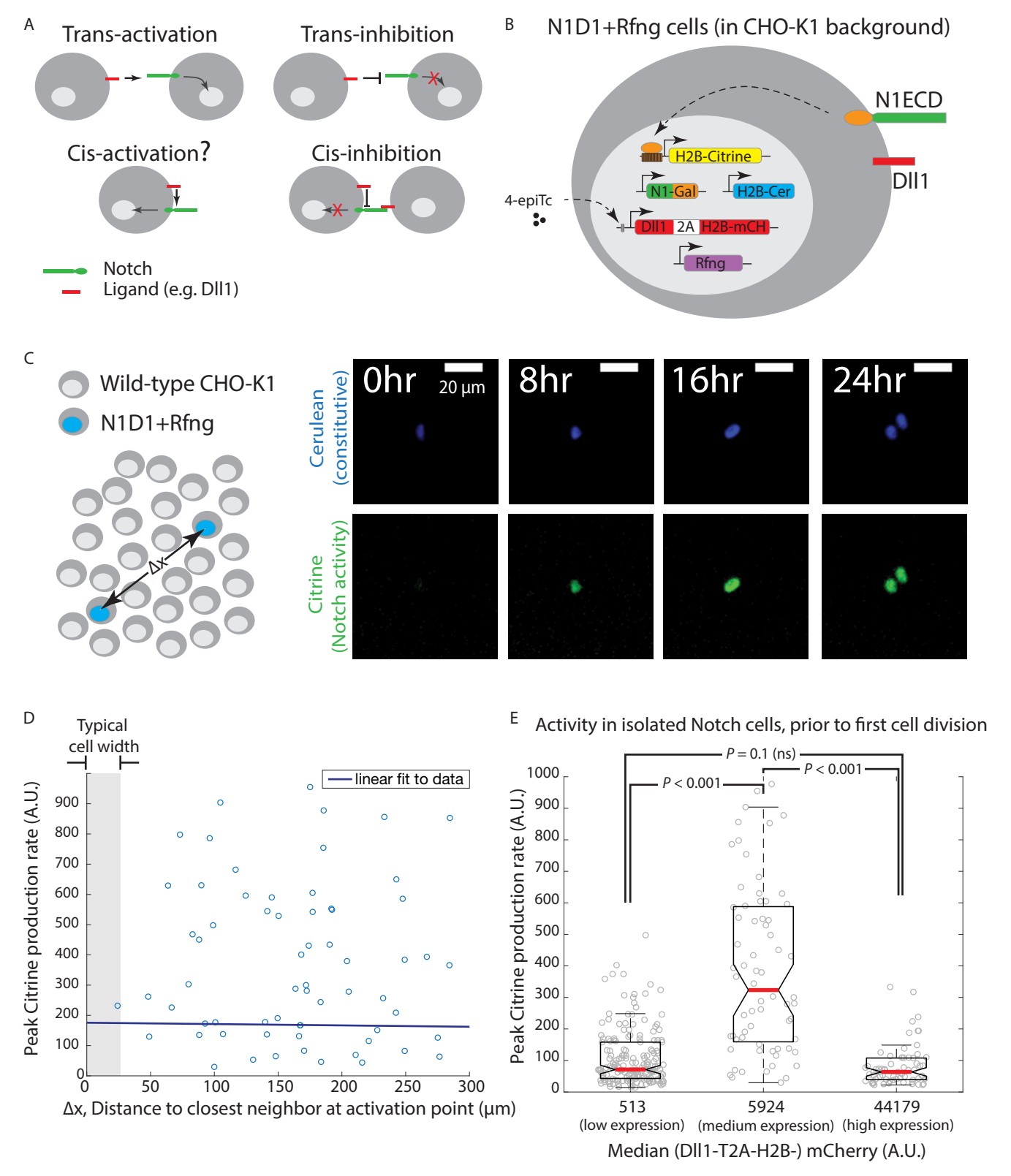

**Figure 1.** Engineered CHO-K1 N1D1 + Rfng cells show ligand-dependent *cis*-activation. (**A**) Schematic of actual and potential *cis*- and *trans*-interaction modes in the Notch pathway. (**B**) Schematic of the N1D1 + Rfng cell line. CHO-K1 cells were engineered to express a chimeric receptor combining the Notch1 extracellular domain ('N1ECD', green) with the Gal4 transcription factor (orange) in place of the endogenous intracellular domain. When

*Figure 1 continued on next page*

*Figure 1 continued*

activated, released Gal4 activates a stably integrated fluorescent H2B-Citrine reporter gene (yellow) through UAS sites (brown) on the promoter. Cells also contain a stably integrated construct expressing Dll1 (red) with a co-translational (2A, white) H2B-mCherry readout ('mCH', red), from a 4-epiTc-inducible promoter. Cells also constitutively express Rfng (purple) and H2B-Cerulean ('H2B-Cer', blue). (C) (*Left*) Schematic of *cis*-activation assay conditions. A minority of N1D1 + Rfng (blue nuclei) cells were mixed with an excess of wild-type CHO-K1 cells (white nuclei). The typical distance between N1D1 + Rfng cells is Δx. (*Right*) Filmstrip showing activation (Citrine fluorescence, green) of an isolated N1D1 + Rfng cell using time-lapse microscopy. Constitutive cerulean fluorescence (blue) in the same cell nucleus is also shown (see *Video 1* for additional examples). (D) Peak Notch activation rate in isolated N1D1 + Rfng cells (y-axis) versus distance to each of its closest neighboring N1D1 + Rfng cell (x-axis) at the point of maximum activity. One cell width is indicated by gray shaded area. Solid blue line indicates linear fit, whose flat slope suggests a cell-autonomous, distance-independent process. (E) Box plots showing the distribution of peak Notch activation rates in isolated N1D1 + Rfng cells prior to the first cell division in the *cis*-activation assay, for three different median Dll1 induction levels (indicated by numbers below bars; see *Figure 1—figure supplement 2A* for corresponding distributions).

DOI: https://doi.org/10.7554/eLife.37880.002

The following figure supplements are available for figure 1:

**Figure supplement 1.** *Cis*-activation assay enables isolation of individual engineered cells.

DOI: https://doi.org/10.7554/eLife.37880.003

**Figure supplement 2.** *Cis*- and *trans*-activation share similar features.

DOI: https://doi.org/10.7554/eLife.37880.004

**Figure supplement 3.** *Cis*-activation occurs with the wild-type Notch1 receptor and in multiple cell types.

DOI: https://doi.org/10.7554/eLife.37880.005

**Figure supplement 4.** *Cis*-activation occurs with endogenous ligands and receptors in Caco-2 and NMuMG cells.

DOI: https://doi.org/10.7554/eLife.37880.006

Here, we used single cell imaging to investigate activation in isolated cells. We find that *cis*-activation is a pervasive property of the Notch signaling pathway. It occurs for multiple ligands (Dll1, Dll4 and Jag1) and receptors (Notch1 and Notch2), and in diverse cell types, including fibroblastic CHO-K1 cells, epithelial NMuMG and Caco-2 cells, and in neural stem cells. *Cis*-activation resembles *trans*-activation in its dependence of signaling response on ligand levels, modulation by R-Fringe, and susceptibility to *cis*-inhibition at high ligand concentrations. Furthermore, *cis*-activation appears to impact the survival of neural stem cells. Finally, mathematical modeling shows that *cis*-activation could expand the capabilities of the Notch pathway, potentially enabling 'negative' Notch signaling and integration of information about levels of *cis*- and *trans*-ligand. Together, these results extend the range of Notch signaling modes and provoke new questions about how *cis*-activation could function in diverse processes.

## Results

### Notch1-Dll1 cells show ligand-dependent *cis*-activation

To analyze *cis*-activation, we sought to develop a synthetic platform that could allow tuning of Notch pathway components and quantitative single-cell read-out of pathway activation (*Figure 1B*). We used the CHO-K1 cell line, which does not naturally express Notch receptors or ligands and has been used in previous studies of the Notch pathway (*Sprinzak et al., 2010*; *LeBon et al., 2014*; *Nandagopal et al., 2018*). We engineered these cells to co-express the Notch ligand Dll1, a chimeric Notch1ECD-Gal4 receptor (N1ECD-Gal4), as well as the Gal4-activated H2B-Citrine fluorescent reporter gene that enables readout of Notch activation (Materials and methods). In these engineered cell lines, receptors are expressed constitutively. Dll1 expression can be induced using the small molecule 4-epi-Tetracycline (4-epiTc) in a dose-dependent manner, and monitored using a co-translational H2B-mCherry fluorescent protein (*LeBon et al., 2014*). Upon activation by Notch ligand, the chimeric N1ECD-Gal4 releases Gal4, which can travel to the nucleus and activate H2B-Citrine expression. Engineered cells also express a Radical Fringe (Rfng) gene, which enhances Notch1-Dll1 interactions through receptor glycosylation (*Moloney et al., 2000*). Finally, these 'N1D1 + Rfng' cells constitutively express nuclear-localized H2B-Cerulean fluorescent protein, which enables their identification in co-culture assays and time-lapse microscopy.

To discriminate *cis*-activation from *trans*-activation, we isolated individual N1D1 + Rfng cells by co-culturing a minority of N1D1 + Rfng cells (1%) with an excess of wild-type CHO-K1 cells ('*cis*-

activation assay', **Figure 1C**, left). We first verified that their relative density was low enough to prevent *trans*-interactions between them, by confirming that a similar fraction of pure receiver cells, which express Notch1 but no ligands, were not activated by N1D1 + Rfng cells (**Figure 1—figure supplement 1**). We then used time-lapse microscopy to measure Notch activity in N1D1 + Rfng cells in the *cis*-activation assay (Materials and methods). At intermediate Dll1 expression levels (with 80 ng/ml 4-epiTc), isolated N1D1 + Rfng cells showed clear activation (**Figure 1C**, right; **Video 1**). As expected for a cell-autonomous process, Notch activity, estimated by the peak rate of Citrine production, was uncorrelated with proximity to neighboring N1D1 + Rfng cells (**Figure 1D**). However, the activity depended strongly on ligand expression levels (**Figure 1E**). Interestingly, this dependence was non-monotonic, peaking at intermediate levels of Dll1 induction, but returning to baseline at high ligand levels (**Figure 1E**; Dll1 induction levels shown in **Figure 1—figure supplement 2A**). This suppression of Notch activity is consistent with the previously described phenomenon of *cis*-inhibition (**del Álamo et al., 2011**; **Sprinzak et al., 2010**; **del Álamo et al., 2011**). We confirmed that the same behavior could be observed in N1D1 + Rfng cells plated sparsely without surrounding wild-type CHO-K1 cells, suggesting that the phenomenon does not depend on the overall cell density (**Figure 1—figure supplement 2B**). Together, these results suggest that Notch1 can be activated by intermediate concentrations of *cis*-Dll1, but that this *cis*-activation is dominated or replaced by *cis*-inhibition at high ligand concentrations.

We next asked how the strength of *cis*-activation compared to that of *trans*-activation, by analyzing the effect of intercellular contact on signaling levels. To control intercellular contact, we varied the fraction (relative density) of N1D1 + Rfng cells in the co-culture, using wild-type CHO-K1 cells to maintain a constant total cell density. In order to increase the throughput of the experiment, we used flow cytometry to measure activation levels after 24 hr of culture (Materials and methods). Total activation levels, which reflect a combination of *cis*- and *trans*-signaling, displayed a non-monotonic dependence on ligand expression for all N1D1 + Rfng fractions, similar to *cis*-activation alone (**Figure 1—figure supplement 2C**, cf. **Figure 1E**). The peak amplitude of total activation was ~ 3 fold higher than *cis*-activation at high N1D1 + Rfng cell densities, but *cis*- and total signaling peaked at the same ligand concentration (**Figure 1—figure supplement 2D**). These results are consistent with overall Notch activation reflecting contributions from both *cis*- and *trans*-interactions, both of which depend similarly on ligand concentration.

In principle, *cis*-activation could be an artifact of the chimeric Notch1ECD-Gal4 receptor. To test this possibility, we analyzed cells co-expressing Dll1 and the wild-type Notch1 receptor (N1$^{WT}$). For readout, we used a previously characterized 12xCSL-H2B-Citrine reporter gene, which can be activated by cleaved NICD through multimerized CSL binding sites in the promoter region (**Figure 1—figure supplement 3A**, left panel) (**Sprinzak et al., 2010**). In the *cis*-activation assay, these 'N1$^{WT}$D1+Rfng' cells showed *cis*-activation and non-monotonic dependence on ligand levels, similar to the responses described above for the N1ECD-Gal4 cells (**Figure 1—figure supplement 3A**, right panel). These results indicate that *cis*-activation occurs for wild-type as well as engineered receptors.

Next, we asked whether *cis*-activation occurs in cell types other than CHO-K1. We analyzed the polarized mammary epithelial cell line NMuMG (**Owens et al., 1974**), which normally express the receptor Notch2 (N2) and ligand Jagged1 (J1) in addition to lower levels of Notch1 (**Figure 1—figure supplement 3B**, left). We first asked whether the components analyzed previously, Notch1 and Dll1, display similar *cis*-activation behavior in this cell type. We therefore deleted endogenous Notch2 and Jagged1 using CRISPR-Cas9 ('NMuMG-ΔN2ΔJ1', Materials and methods, **Figure 1—figure supplement 3B**, middle) and added inducible Dll1,

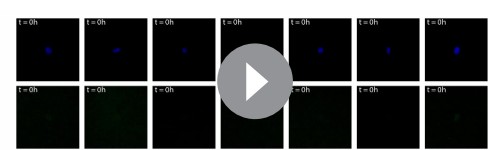

**Video 1.** *Cis*-activation of isolated engineered CHO-K1 cells. Examples of isolated CHO-K1 N1D1 + Rfng cells activating prior to cell division in the *cis*-activation assay. (*Top row*) Blue channel shows fluorescence of the constitutively expressed nuclear H2B-Cerulean protein. (*Bottom row*) Green channel shows fluorescence of the Notch-activated H2B-Citrine reporter protein (also nuclear). The same intensity scales have been applied to each frame of the movie and for all cells. Interval between individual frames of the movie is 30 min. Non-fluorescent CHO-K1 cells surround each isolated fluorescent cell but are not visible.

DOI: https://doi.org/10.7554/eLife.37880.007

constitutive Rfng, and the chimeric Notch1 receptor-based reporter system described above (*Figure 1—figure supplement 3C*). In addition to providing a cleaner background, the deletion of endogenous N2 and J1 also enhanced the response of N1ECD-Gal4 to Dll1 (*Figure 1—figure supplement 3B*, right), possibly by eliminating competition of the ectopic components with the endogenous Notch components. Since Notch signaling in polarized epithelial cells relies on the proper apical localization of Notch ligands and receptors (*Sasaki et al., 2007*), and localization is often controlled through interactions occurring in the intracellular domain (*Benhra et al., 2010*; *Benhra et al., 2011*), we analyzed N1ECD-Gal4 receptors fused to different parts of the intracellular domain. We discovered that attachment of the ankyrin (ANK) domain ('N1ECD-Gal4-ANK') improved apical localization of the receptor and further enhanced signaling levels (*Figure 1—figure supplement 3D*, Materials and methods). Note that this modification appeared to be unnecessary in CHO-K1 fibroblasts, where the N1ECD-Gal4 showed similar surface localization as the full-length N1 (N1$^{wt}$, *Figure 1—figure supplement 3E*). When these 'NMuMG N1D1 + Rfng' cells were analyzed using time-lapse microscopy in the *cis*-activation assay, isolated cells showed clear activation (*Figure 1—figure supplement 3F*, *Video 2*). This activation displayed a non-monotonic dependence on Dll1 expression (*Figure 1—figure supplement 3G*), similar to CHO-K1 cells, indicating that the *cis*-activation phenomenon could be general to multiple cell types.

To further examine *cis*-activation, we next asked whether *cis*-activation occurs with the endogenously expressed components (Notch2 and Jagged1) in NMuMG cells. To test this, wild-type NMuMG cells, pre-incubated with DAPT, were plated sparsely with or without continued DAPT treatment (see Materials and methods). 6 hr later, the expression levels of the Notch target gene Hes1 were analyzed using qRT-PCR. Cells removed from DAPT upregulated Hes1 levels compared to cells that remained in DAPT, and addition of ectopic Dll1 to the cells further increased Hes1 upregulation (*Figure 1—figure supplement 4B*). To verify that Hes1 upregulation occurred in isolated cells, we used single-molecule HCR-FISH (*Choi et al., 2010*; *Choi et al., 2018*) to detect Hes1 mRNA transcripts at the single-cell level. Similar to the bulk qRT-PCR results, this analysis showed that Hes1 was modestly upregulated in isolated wild-type NMuMG cells by 6 hr after DAPT removal (*Figure 1—figure supplement 4C*). Again, Hes1 induction levels could be increased by the addition of exogenous Dll1 (*Figure 1—figure supplement 4C and D*). These results are consistent with the conclusion that *cis*-activation occurs endogenously in NMuMG cells.

*Cis*-activation also occurred in the human colorectal adenocarcinoma cell line Caco-2, where Notch signaling is known to regulate proliferation and differentiation (*Sääf et al., 2007*; *Dahan et al., 2011*). To measure endogenous Notch activity, we transfected Caco-2 cells with the 12xCSL-H2B-Citrine reporter construct (used in *Figure 1—figure supplement 3A*). To analyze *cis*-activation, we plated the transfected cells sparsely, with or without the Notch inhibitor DAPT (*Dovey et al., 2001*) (see Materials and methods). After 24 hr, DAPT-treated cells displayed lower levels of Notch activation compared to untreated cells, consistent with *cis*-activation by endogenous Notch components (*Figure 1—figure supplement 4A*).

Taken together, these results demonstrate that *cis*-activation is a general aspect of Notch signaling, occurring in diverse cell types and with endogenous Notch receptors and ligands. In cells co-expressing Notch1, Dll1, and R-Fringe, *cis*-activation strength depends on ligand concentration. In CHO and NMuMG cells, *cis*-activation peaks at intermediate ligand concentrations, replaced by *cis*-inhibition at the highest ligand levels (*Figure 1—figure supplement 2* and *Figure 1—figure supplement 3G*).



**Video 2.** *Cis*-activation of isolated engineered NMuMG cells. Examples of isolated NMuMG N1D1 + Rfng cells activating prior to cell division in the *cis*-activation assay. (*Top row*) Blue channel shows fluorescence of the constitutively expressed nuclear H2B-Cerulean protein. (*Bottom row*) Green channel shows fluorescence of the Notch-activated H2B-Citrine reporter protein (also nuclear). The fluorescence image is overlaid on the DIC image (grey), in which surrounding non-fluorescent NMuMG cells can be seen. The same intensity scales have been applied to each frame of the movie and for all cells. Interval between individual frames of the movie is 30 min.
DOI: https://doi.org/10.7554/eLife.37880.008

## *Cis*-activation changes with ligand-receptor affinity

Ligand-receptor interaction affinities differ across ligand-receptor pairs, and can be modulated by glycosyltransferases like Rfng (*Moloney et al., 2000*; *Yang et al., 2005*; *Taylor et al., 2014*). Rfng is known to increase *trans* Notch1-Dll1 signaling. To understand how it affects *cis*-activation, we compared the N1D1 + Rfng line to its parental line ('N1D1'), lacking expression of Rfng. N1D1 cells showed ligand-dependent *cis*-activation, but at reduced levels (*Figure 2A*). As with N1D1 + Rfng, *cis*-activation dominated at intermediate Dll1 concentrations, while *cis*-inhibition dominated at high Dll1 concentrations. Further, extending the analysis of Notch activation to conditions with increased intercellular contact, we observed a similar dependence on cell fraction and Dll1 expression with and without Rfng; the two states differed in signal amplitude but not the shape of the ligand response (*Figure 2B*). Thus, Rfng increases the amplitude of both *cis* and *trans* signaling without affecting the overall dependence of signaling on Dll1 expression level.

We next analyzed how identity of the ligand affects *cis*-activation. Compared to Dll1, the ligand Dll4 has increased affinity for Notch1 (*Andrawes et al., 2013*). We engineered CHO-K1 cells to stably express either an inducible Dll4-T2A-H2B-mCherry or Dll1-T2A-H2B-mCherry, along with a constitutive Notch1ECD-Gal4 Notch reporter system. To enable direct comparison, we performed the *cis*-activation analysis on polyclonal populations for the two cell lines. Compared to the Dll1-expressing cells, Dll4-expressing cells showed enhanced *cis*-activation and *cis*-inhibition, exhibiting greater peak reporter activity at intermediate ligand expression levels but comparable activity at the highest ligand expression levels (*Figure 2C*). Consistent with previous studies showing that Rfng does not increase Dll4-Notch1 affinity (*Taylor et al., 2014*), expressing Rfng in the N1D4 cells did not further increase *cis*-activation or *cis*-inhibition (*Figure 2—figure supplement 1A*). Taken together, these data suggest that stronger ligand-receptor interactions, either through Rfng or through a higher affinity Notch ligand like Dll4, enhance both *cis*-activation and *cis*-inhibition.

## Notch2 shows stronger *cis*-activation but decreased *cis*-inhibition compared to Notch1

To investigate *cis*-activation with other Notch receptors, we engineered CHO-K1 cells to express a Notch2 reporter system (N2ECD-Gal4) along with inducible Dll1- or Dll4-T2A-H2B-mCherry, as described previously. Both N2D1 and N2D4 cell populations showed ~ 3 fold higher maximal *cis*-activation compared to their Notch1 counterparts (*Figure 2D*, note difference in scale compared to *Figure 2C*). Moreover, unlike Notch1, Notch2 showed similar levels of *cis*-activation by the Dll1 and Dll4 ligands (*Figure 2D*). Strikingly, the profile of activation was monotonic, with *cis*-activation persisting even at the highest ligand levels tested (*Figure 2—figure supplement 2*). Together, these results indicate that Notch2 undergoes *cis*-activation, does so at a higher level than Notch1, and is not *cis*-inhibited as strongly as Notch1.

## *Cis*-activation affects neural stem cell maintenance

To test whether *cis*-activation could impact Notch-mediated cellular behaviors, we analyzed mouse cortical neural stem cells (NSCs), in which Notch signaling regulates self-renewal and differentiation (*Bertrand et al., 2002*; *Kageyama et al., 2008*). Primary NSCs can be cultured and propagated in vitro under defined media conditions and cell density (*Daadi, 2002*). Bulk RNA sequencing revealed that these cells express high levels of Notch1, Dll1, and Lfng, and lower levels of Notch2 and Rfng, suggesting that NSCs have the potential to *cis*-activate (*Figure 3—figure supplement 1A*, Materials and methods).

To identify suitable gene targets for assaying Notch activation, we next analyzed the expression of the *Hes/Hey* genes, with or without the Notch inhibitor DAPT for 12 hr. Since NSC culture conditions include treatment with the EGF and FGF growth factors, and there is evidence for crosstalk between the growth factors and Notch signaling pathways in these cells (*Aguirre et al., 2010*); *Nagao et al., 2007*), we compared Notch activation with or without the Notch inhibitor DAPT (10 µM), under standard (20 ng/ml EGF, 20 ng/ml FGF) and reduced (0.5 ng/ml EGF, no FGF) growth factor conditions (Materials and methods). Canonical Notch target genes *Hes1*, *Hes5*, and *Hey1* decreased in response to DAPT, and did so more strongly at reduced growth factor concentrations (*Figure 3—figure supplement 1B*).

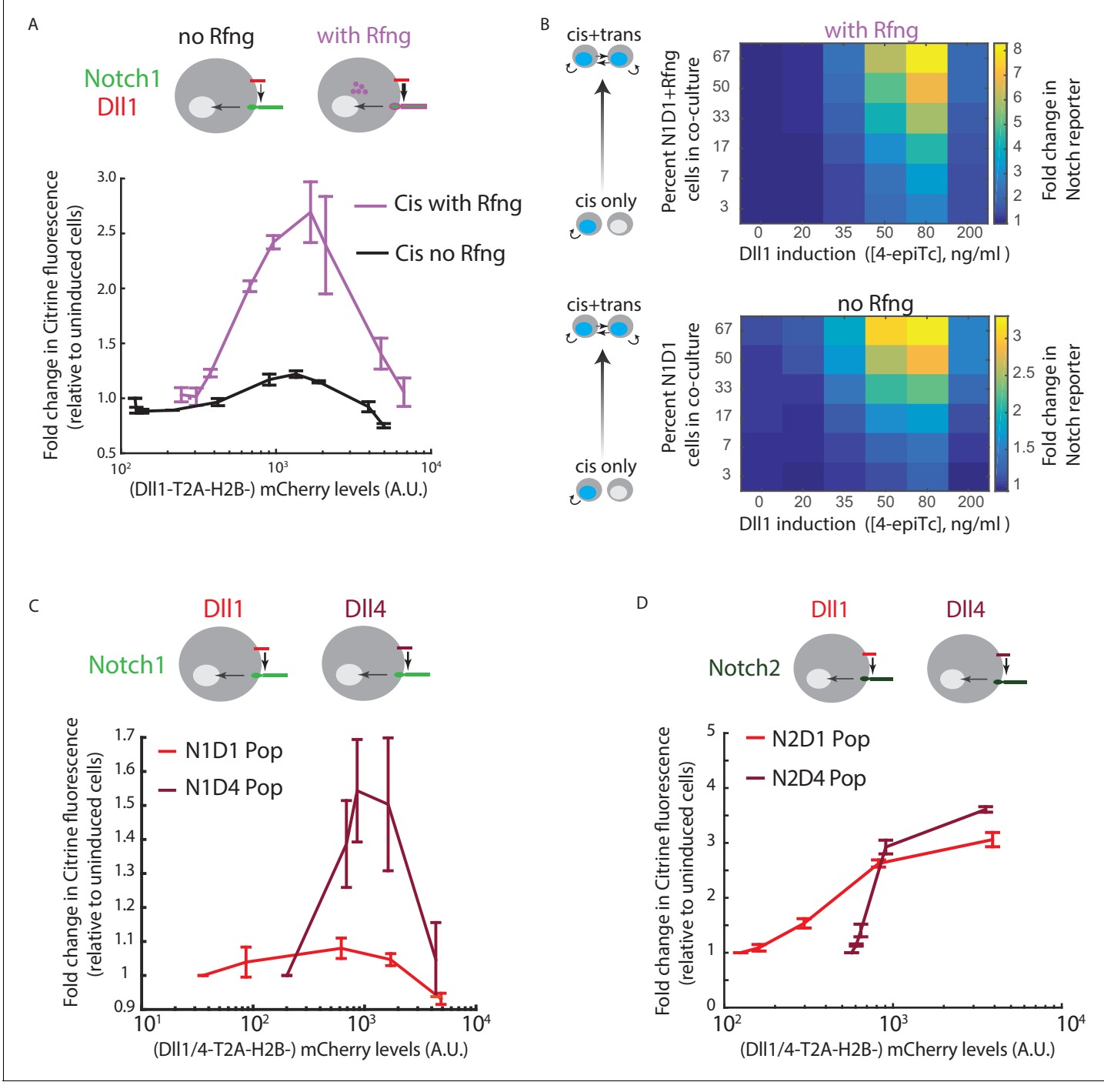

**Figure 2.** *Cis*-activation is affected by changes in ligand-receptor affinity. (**A**) (*Top*) Cell lines used for analyzing effect of Rfng on *cis*-activation. (*Bottom*) Plots showing mean Notch activation (reporter Citrine fluorescence normalized to background fluorescence in uninduced cells) in N1D1 (black) or N1D1 + Rfng (purple) cells expressing different levels of Dll1 (measured using co-translational mCherry fluorescence). Error bars indicate s.e.m (n = 3 replicate experiments). (**B**) Heatmaps of mean Notch activation (n = 3 replicates), relative to background reporter fluorescence, in N1D1 + Rfng (upper panel) or N1D1 (lower panel) cells induced with different [4epi-Tc] (columns) and cultured at different relative fractions (rows). Upper panel is the same data in *Figure 1—figure supplement 2C*, replotted for direct comparison. Rfng expression predominantly affects signal amplitude (compare intensity scales). (**C,D**) (*Top*) Cell lines used for analyzing effect of ligand on *cis*-activation of Notch1 (**C**) or Notch2 (**D**). (*Bottom*) Comparison of mean *cis*-activation in polyclonal populations ('Pop') of cells co-expressing Dll1 or the higher affinity ligand Dll4 with the indicated receptor, as a function of ligand expression, read out by co-translated H2B-mCherry fluorescence. Values represent mean of 3 replicates. Error bars indicate s.e.m. Note difference in y-axis scales between panels C and D.

*Figure 2 continued on next page*

*Figure 2 continued*

DOI: https://doi.org/10.7554/eLife.37880.009

The following figure supplements are available for figure 2:

**Figure supplement 1.** Rfng does not modify the *cis*-activation behavior of N1D4 cells.

DOI: https://doi.org/10.7554/eLife.37880.010

**Figure supplement 2.** Notch2 lacks *cis*-inhibition with Dll1 or Dll4.

DOI: https://doi.org/10.7554/eLife.37880.011

To analyze *cis*-activation in NSCs, we plated cells at low density in reduced growth factor conditions (0.1 ng/ml EGF, no FGF), and cultured them with or without 10 µM DAPT (*Figure 3A*, Materials and methods). After 6 hr, we assayed mRNA transcript levels of Hes1, Hey1, and Hes5 in isolated cells using single-molecule HCR-FISH (*Choi et al., 2010*; *Choi et al., 2018*) (*Figure 3B*, *Figure 3— figure supplement 1C*). DAPT treatment decreased Hes1, Hes5, and Hey1 by mean fold changes of 2.5 (95% confidence interval, 2.1–4.1), 1.9 (1.3–2.5), and 1.2 (1.1, 1.2), respectively, consistent with *cis*-activation of Notch target genes.

We next asked whether *cis*-activation could potentially affect the Notch-dependent process of NSC maintenance. We treated cells with Dll1-targeting siRNA or control siRNA for 48 hr and then plated them at low density in low growth factor conditions (see Materials and methods). We also examined the effect of more complete Notch inhibition through DAPT treatment on the control siRNA-treated samples.

Under these conditions, DAPT treatment strongly decreased the number of cells after 24 hr, consistent with a role for *cis*-activation in promoting cell survival (*Figure 3C*, right panel). (Note, however, that we cannot rule out Notch-independent contributions of γ-secretase inhibition by DAPT). Dll1 siRNA treatment also decreased survival, albeit more weakly, likely due to the incomplete knockdown of Dll1 protein levels (*Figure 3C*, right panel and *Figure 3D*, see Materials and methods). Survival remained unaffected under high growth factor conditions, suggesting the effect was not due to general cellular toxicity caused by siRNA treatment (*Figure 3C*, left panel). The impact of Notch pathway downregulation on cell survival could be reversed by plating isolated cells (without siRNA treatment) on recombinant Dll1ext-IgG ligands ('Dplate'). This condition produced a striking increase in cell numbers at 24 hr (*Figure 3E*, Materials and methods). These results show that perturbation of Notch signaling in both directions produces corresponding changes in survival and suggest that Notch *cis*-activation can affect initial NSC survival in low growth factor conditions.

## *Cis*-activation requires cell surface interactions between ligands and receptors

To gain insight into where *cis*-activation occurs in the cell, we tested whether cell surface ligand-receptor interactions were required for productive signaling (*Figure 4A*). Treatment of cells with soluble recombinant N1ECD-Fc (rN1ECD-Fc) receptors has been shown to prevent *trans*-signaling by blocking surface ligands (*Klose et al., 2015*). We first confirmed that activation levels in densely-plated N1D1 + Rfng cells decreased when they were incubated in rN1ECD-Fc containing media for 24 hr, compared to IgG-treated controls (*Figure 4—figure supplement 1A*). Interestingly, a similar decrease in activation levels could be observed in N1D1 + Rfng cells plated in the *cis*-activation assay (*Figure 4B*), suggesting that blocking ligand-receptor interactions at the surface reduces *cis*-activation. This effect was not limited to soluble receptor fragments; co-culturing a minority (5%) of N1D1 + Rfng cells with an excess of Notch1-only expressing cells similarly reduced *cis*-activation, and by a comparable amount (*Figure 4—figure supplement 1B*). We further perturbed cell-surface ligand-receptor interactions by treating cells with Blebbistatin, an inhibitor of non-muscle myosin II, known to disrupt cellular adhesion and protrusions (Materials and methods) (*Liu et al., 2010*; *Shutova et al., 2012*). Similar to rN1ECD-Fc, treatment with Blebbistatin decreased both *cis*- and *trans*-activation of N1D1 + Rfng cells to similar extents (*Figure 4—figure supplement 1C*). Together, these results suggest that *cis*-activation requires surface presentation of the ligand, and are consistent with a model in which productive *cis* ligand-receptor interactions, like *trans*

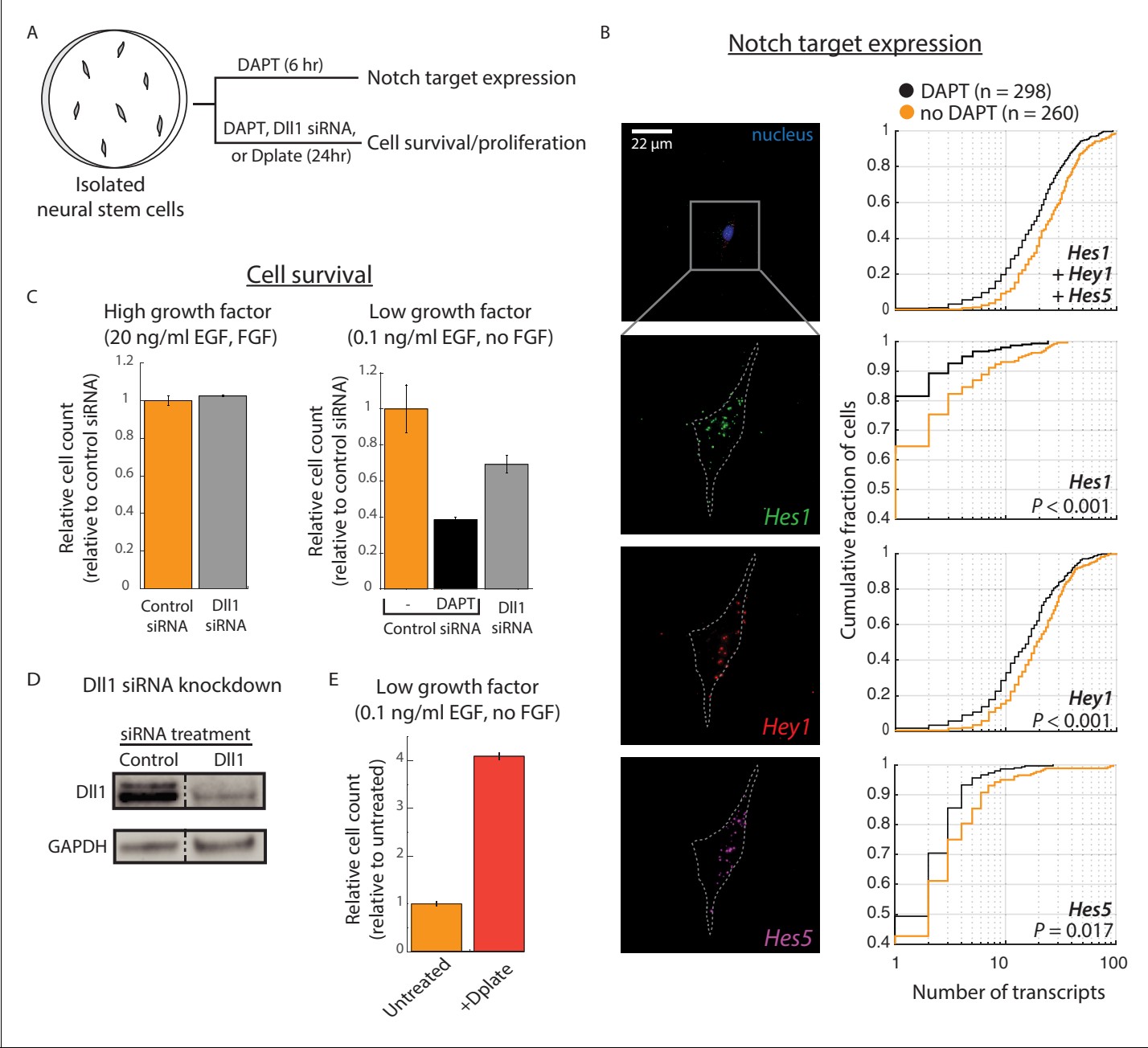

**Figure 3.** *Cis*-activation occurs in neural stem cells and regulates survival. (**A**) E14.5 mouse cortical neural stem cells (NSCs) were plated sparsely and treated with ± 10 µM DAPT, cultured under growth conditions with low growth factors, and subsequently assayed for expression of Notch target genes and cell survival. (**B**) (*Left*) Representative example of an isolated NSC (top panel, DAPI-stained nucleus shown; note lack of neighboring cells within ~ 50 µm) not treated with DAPT, assayed for expression of Hes1 (green), Hey1 (red), and Hes5 (magenta) mRNA using multiplexed single-molecule HCR-FISH (see Materials and methods). (*Right*) Cumulative distribution plots of gene expression in DAPT-treated (black) and untreated (orange) cells (see Materials and methods for transcript quantification). *P*-values calculated using two-sided KS-test. See *Figure 3—figure supplement 1C* for additional examples of isolated cells showing Hes/Hey expression. (**C**) (*Left*) Median of relative cell number in samples of isolated cells that were pre-treated with control siRNA (orange, see Materials and methods) or Dll1-targeting siRNA (grey) and cultured in complete growth medium for 24 hr. (*Right*) Median of relative cell number when the same siRNA-treated cells were cultured in low growth factor medium for 24 hr. Control siRNA treated cells were additionally treated with DAPT for this period. Error bars represent s.e.m, n = 2 biological replicates (**D**) Western blot analysis of endogenous Dll1 protein after siRNA mediated knockdown in NSCs. Note that Dll1 protein is reduced, but not eliminated, in NSC cells treated with Dll1-targeting siRNA compared to cells treated with control siRNA. GAPDH detection was used to assess protein loading. Dashed line indicates lanes spliced together from a single protein gel. (**E**) Median of relative cell number after 24 hr in samples of isolated cells plated on normal culture surfaces (orange)
*Figure 3 continued on next page*

*Figure 3 continued*
or on surfaces coated with recombinant Dll1ext-IgG protein ('Dplate', red, see Materials and methods). Cells were cultured in low growth factor conditions. Error bars represent s.e.m, n = 4 biological replicates.
DOI: https://doi.org/10.7554/eLife.37880.012
The following figure supplement is available for figure 3:
**Figure supplement 1.** RNAseq analysis of Notch pathway component expression in neural stem cells.
DOI: https://doi.org/10.7554/eLife.37880.013

interactions, occur at the cell surface. However, a more complete understanding of the *cis*-activation mechanism will require additional analysis.

We next asked whether *cis*-activation requires interactions with the culture dish surface. For example, *cis*-activation could involve focal adhesions formed at points of contact with the dish. Alternatively, it could involve cells depositing ligands on the culture surface, which *trans*-activate cell-surface receptors. To address this question, we analyzed N1D1 + Rfng *cis*-activation in a suspension culture.

To create suspension cultures, cell adhesion to the plate surface was prevented by pre-coating the surface with silicone (*Nienow et al., 2016*) and putting the plate on a rocker for the duration of the experiment (*Figure 4C*, Materials and methods). Under these conditions, N1D1 + Rfng cells (co-cultured with an excess of wild-type CHO-K1 cells as in the *cis*-activation assay, see *Figure 1C*) continued to show *cis*-activation (*Figure 4D*). We note that culturing cells in suspension caused slight reductions in cell-surface Notch1 and Dll1 levels (*Figure 4—figure supplement 1D*), which could account for the minor reduction in peak *cis*-activation observed in suspension cells compared to adherent cells. Nevertheless, these results suggest that *cis*-activation is a cell-autonomous process that does not require extensive interactions with the culture surface.

Together, our results support a model in which *cis*-activation arises in a cell-autonomous manner from interactions between ligands and receptors on the cell surface. More generally, the observed similarities between *cis*- and *trans*-activation in their dependence on ligand concentration and ligand-receptor affinities, and sensitivity to perturbations, could reflect a common underlying mechanism of activation.

## *Cis*-activation enables integration of intra- and extracellular information and negative signaling

To understand what underlying interactions could explain key features of *cis*-activation, we developed a series of simplified mathematical models of Notch-ligand *cis*-interactions at steady-state, and compared their behaviors to experimental observations.

We first considered the simplest case of Notch, denoted N, and Delta, denoted D, reversibly interacting in *cis* to form a single activation-competent complex, denoted $C^+$, which can subsequently undergo cleavage to release NICD (*Figure 5A*, Model 0, Materials and methods). We simulated this model using 10,000 biochemical parameter sets, chosen using Latin Hypercube Sampling to uniformly cover parameter space (*McKay et al., 1979*) (*Figure 5B*, *Figure 5—figure supplement 1*, Materials and methods). For each parameter set, we quantified the degree of non-monotonicity in the concentration of $C^+$ as a function of the Delta production rate, $\alpha_D$ (*Figure 5B,D*). Model 0 did not produce non-monotonic responses (*Figure 5C*, Materials and methods), indicating that *cis*-activation at low ligand concentrations and *cis*-inhibition at high ligand concentrations cannot both result from a single underlying type of *cis*-complex.

We next considered a more complex model in which Notch and Delta can generate two distinct types of *cis*-complexes, $C^+$ and $C^-$, with only the former competent to activate (*Figure 5A*, Model 1, Materials and methods). However, this model was similarly unable to recapitulate non-monotonic $C^+$ behavior (*Figure 5C*, Materials and methods).

We reasoned that a sequential process of *cis*-complex formation could generate active $C^+$ and inactive $C^-$ at lower and higher Delta production rates, respectively (*Figure 5A*). For instance, $C^-$ could be formed from $C^+$ through additional interactions of the active complex with itself ($C^+ + C^+ \rightarrow C^-$, Model 2b), through interactions with free N and D (N + D + $C^+ \rightarrow C^-$, Model 2a), or interactions with ligand (D + $C^+ \rightarrow C^-$, Model 2c). (Note that $C^-$ has different constituents in each of these

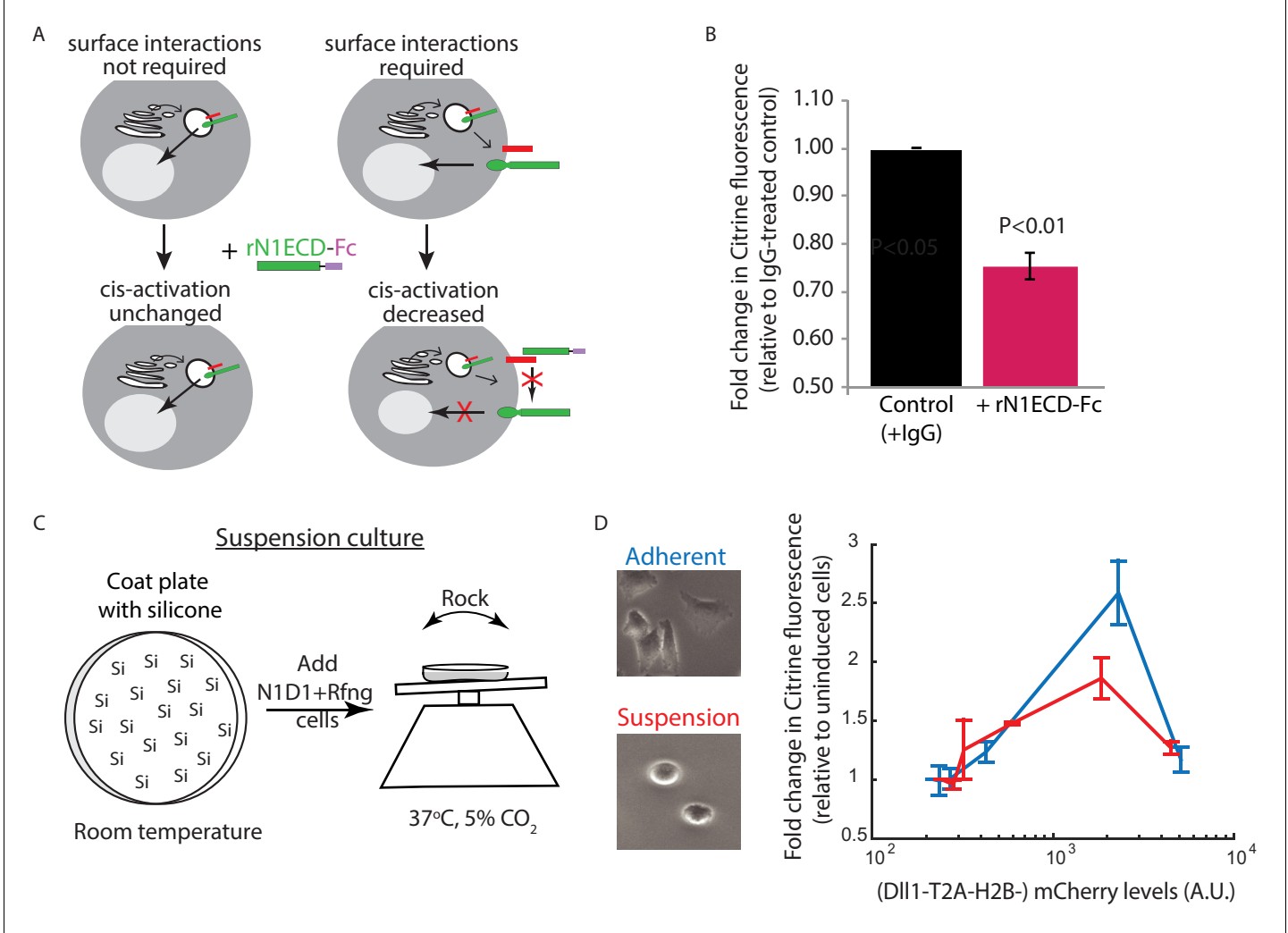

**Figure 4.** Receptor-ligand cell surface interactions are necessary for *cis*-activation. (A) Schematics showing how soluble recombinant N1ECD-Fc protein (rN1ECD-Fc) can be used to test whether surface interactions between ligand (red) and receptor (green) are necessary for *cis*-activation. (*Left*) With intracellular *cis*-activation, addition of extracellular rN1ECD-Fc should not affect *cis*-activation levels. (*Right*) If surface interactions are necessary for *cis*-activation, rN1ECD-Fc treatment should reduce activation levels by competing with receptors for cell-surface ligands. (B) Comparison of mean Notch activation in N1D1 + Rfng cells incubated with rNotch1ECD-Fc receptors (magenta) or an IgG control (black) for 24 hr (see Materials and methods). Cells were plated for a *cis*-activation assay and analyzed by flow cytometry < 24 hr post-plating. Error bars represent s.e.m (n = 3 replicate experiments). *P*-values calculated using the one-sided Student T-test. (C) Schematic of procedure for culturing N1D1 + Rfng cells in suspension. Plates were coated with a Silicone solution ('Si'), and cells were subsequently plated for a *cis*-activation assay (co-culture of 5 × 10³ N1D1 + Rfng + 150 × 10³ CHO-K1 cells). The plate was incubated at 37°C, 5% $CO_2$ on a rocker to prevent cells from adhering to the plate surface. (D) (*Left*) Representative image of cells grown in adherent (blue) or suspension (red) conditions. (*Right*) Comparison of mean Notch activation levels, relative to background reporter fluorescence, in N1D1 + Rfng cells cultured for 24 hr in suspension (red) or in adherent conditions (blue), for different Dll1 expression levels (measured using co-translated mCherry fluorescence). Cells were in *cis*-activation co-culture conditions (5 × 10³ N1D1 + Rfng + 150 × 10³ CHO-K1 cells). Error bars represent s.e.m (n = 3 replicates).

DOI: https://doi.org/10.7554/eLife.37880.014

The following figure supplement is available for figure 4:

**Figure supplement 1.** Surface perturbations affect N1D1+Rfng *cis*-activation.
DOI: https://doi.org/10.7554/eLife.37880.015

models). Alternatively, formation of C⁻ might require higher-order Notch-Delta interactions than required for formation of C⁺. For example, C⁻ could require interaction of 2 ligands with two receptors (2N + 2D → C⁻, Model 2d). In each of these models, the inactive C⁻ complex is formed through increased clustering of ligands and/or receptors compared to the active C⁺ complex. This scheme is

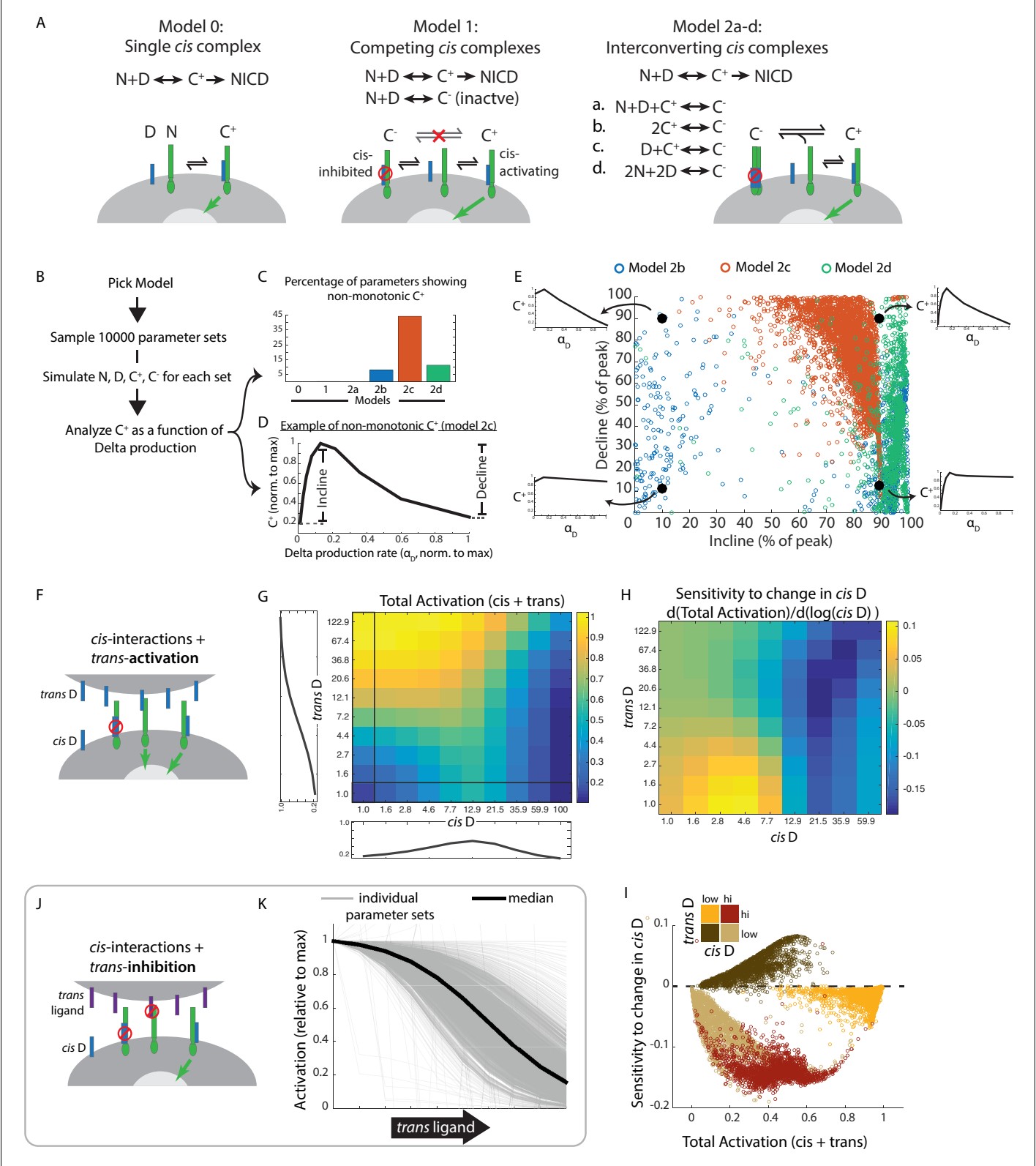

**Figure 5.** Mathematical model of *cis*-activation reveals potential roles in signal processing. (**A**) In each *cis*-activaiton model, Notch ('N', green) and Delta ('D', blue) interact to produce one or more *cis*-complexes, which can be active ('C$^+$'), producing NICD (green arrow) or inhibited ('C$^-$', red circle). In Models 2a-d, C$^+$ is formed through the same interaction, but C$^-$ formation differs for each of the included models. (**B**) Overview of simulations (see Materials and methods). (**C-E**) Results of simulations shown. (**C**) Percentage of parameters that lead to non-monotonic C$^+$ behavior in each of the

*Figure 5 continued on next page*

*Figure 5 continued*

models (see Material and methods for assessment of this feature) (D) Example of non-monotonic dependence of $C^+$ on Delta production rate ('$\alpha_D$'), generated in Model 2 c for one choice of parameter values. The fractional incline and decline features used to characterize the degree of non-monotonicity (and plotted in panel E) are shown. (E) Scatter plot of fractional incline vs. decline for each non-monotonic $C^+$ profile produced by Models 2b-d. Filled black circles and associated schematic plots highlight $C^+$ profile shapes corresponding to different incline vs. decline levels. (F) Schematic of model including both *cis*- and *trans*-activation. Notch receptors can interact with intracellular Delta ('*cis* D') or extracellular Delta ('*trans* D') to form *cis*- and *trans*-complexes, respectively. *Cis*-complexes can be either inhibited (red circle) or activating (green arrow), while *trans* complexes are activating. (G) Example of total activation (levels of activating *cis* + *trans* complexes) as a function of *cis* and *trans* D, for a single set of parameters producing non-monotonic *cis*-activation. (H) Corresponding sensitivity to change in *cis* D for the example in G. This sensitivity ('d(Total Activation)/d(log (*cis* D))') is defined as the change in total activation upon constant fold-changes in *cis* D levels, and is derived from G by computing the difference between adjacent columns of the total activation matrix. (I) Scatter plot showing median values of total activation vs. sensitivity to change in *cis* D in different regimes of *cis* and *trans* D (high *cis*/high *trans* - red, high *cis*/low *trans* - beige, low *cis*/high *trans* - orange, low *cis*/low *trans* - brown). Each circle represents results obtained using a single set of parameters in Model 2c (with *trans*-activation). (J) Schematic of model including *cis*- and *trans*-inhibition. Notch receptors can interact with intracellular Delta ('*cis* D', blue) or extracellular ligand ('*trans* ligand', purple) to form *cis*- and *trans*-complexes, respectively. *Cis*-complexes can be either inhibited (red circle) or activating (green arrow), while *trans* complexes cannot activate. (K) Dependence of total activation levels on *trans*-ligand, for *cis* D production rate corresponding to peak *cis*-activation. Each grey line represents behavior for a single set of parameters, while the black line represents the median response across all tested parameters.
DOI: https://doi.org/10.7554/eLife.37880.016

The following source data and figure supplement are available for figure 5:

**Source data 1.** MATLAB code and parameter sets used for models in *Figure 5*.
DOI: https://doi.org/10.7554/eLife.37880.018

**Figure supplement 1.** 1Latin Hypercube Sampling generates evenly distributed parametersScatter plots showing pairwise distributions of parameters tested in Models 1 and 2a-d.
DOI: https://doi.org/10.7554/eLife.37880.017

broadly consistent with previous observations that Notch ligands and receptors can form clusters (***Bardot et al., 2005***; ***Nichols et al., 2007a***; ***Nandagopal et al., 2018***).

Strikingly, these models produced non-monotonic $C^+$ profiles more frequently than Models 0 and 1, across a similar set of parameter values (***Figure 5C***). For example, Model 2c gave rise to non-monotonic behavior for over 40% of tested parameter sets (***Figure 5C***). Moreover, when we analyzed the shape of the $C^+$ profile (***Figure 5D***), Model 2c came closest to reproducing the experimentally observed non-monotonic shape, frequently showing nearly complete attenuation of activation at the highest Delta production rates (***Figure 5E***, cf. ***Figure 1E***, ***Figure 1—figure supplement 2C***). Other models produced non-monotonic $C^+$ profiles with more modest declines of 30–40% at the highest ligand production rates. While this analysis does not uniquely identify a specific molecular mechanism, it suggests that multiple distinct *cis*-complexes are likely required to explain the observed non-monotonic behavior.

Next, to understand how *cis* and *trans* interactions together determine signaling behavior, we incorporated *trans* interactions in Model 2c. Specifically, we assumed that *trans*-ligands interact with Notch to form productive *trans* complexes, denoted T, and do so with the same rates of formation, dissociation, and degradation as the active *cis*-complexes, $C^+$ (***Figure 5F***). For each non-monotonic parameter set in ***Figure 5E***, we quantified the total concentration of active complexes (T + $C^+$) across a range of *trans*-Delta levels and *cis*-Delta production rates (see Materials and methods).

Using this model, we first asked whether *cis*-activation could enable cells to distinguish between their own ligand levels and that of their neighbors. Multiple combinations of *cis*-Delta and *trans*-Delta levels produced the same level of signaling (***Figure 5G***), indicating that the cell cannot in general infer *cis* and *trans* Delta levels based solely on total Notch activity. However, the sensitivity of Notch activity to *changes* in *cis*-Delta differed strongly between otherwise similar signaling regimes, as illustrated for a single parameter set in ***Figure 5H***. This could also be observed more generally across parameter sets (***Figure 5I***). This analysis suggests that by dynamically modulating its own ligand expression, a cell could, in principle, use *cis*-activation to compare *cis* and *trans* ligand levels.

With the same model, we next explored ways in which inhibitory *trans* ligands could combine with *cis*-activation to produce new modes of signaling. For example, Jagged1 forms inactive *trans* complexes with Notch receptors glycosylated by Lfng (***Shimizu et al., 2001***; ***Moloney et al., 2000***; ***LeBon et al., 2014***). To represent this type of *trans* interaction, we incorporated an inactive *trans* complex, $T^-$, in Model 2c, and analyzed the dependence of Notch activity on the concentration of

*trans* ligand (*Figure 5J,K*, Materials and methods). At *cis*-ligand levels that produce peak *cis*-activation, the inhibitory *trans*-ligand decreased Notch activity in a dose-dependent fashion across multiple parameter sets (*Figure 5K*). This effect resulted from *trans*-ligands effectively competing with *cis*-ligands for a common pool of Notch receptors. In this way, *cis*-activation could enable a 'negative' mode of intercellular Notch signaling, complementing the standard activating mode, much as repression complements activation in gene regulation.

## Discussion

*Cis*-activation is an example of autocrine signaling, which occurs in cytokine, Wnt, BMP, and other signaling pathways (*Fang et al., 2013*; *Feinerman et al., 2010*; *Babb et al., 2017*; *Shukunami et al., 2000*; *Yokoyama et al., 2017*). Typically, autocrine signaling occurs when molecules (e.g. hormones) released from a cell bind to and activate receptors on the cell from which they were synthesized (*Leibiger et al., 2012*). However, membrane-anchored molecules can also produce autocrine signaling. For example, *cis*-activation by cell adhesion molecules (CAMs) helps induce neurite outgrowth during neuronal development (*Sonderegger and Rathjen, 1992*).

Autocrine signaling has been less explored in the Notch pathway (*Formosa-Jordan and Ibañes, 2014a*; *Hsieh and Lo, 2012*). One reason may be that the well-known phenomenon of *cis*-inhibition appears to rule out *cis*-activation (*Sprinzak et al., 2010*; *LeBon et al., 2014*; *Miller et al., 2009*). Additionally, it can be difficult to disentangle *cis*- and *trans*-activation in the context of a tissue or an in vitro system where *cis*-ligands and *trans*-ligands are simultaneously present (*Eddison et al., 2000*; *Hartman et al., 2010*; *Daudet and Lewis, 2005*; *Sprinzak et al., 2010*).

To address these issues, we used an in vitro system that allows tunable control of ligand expression and readout of Notch activity in individual cells, across different cell densities (*Sprinzak et al., 2010*). This system enabled analysis of signaling by ligand- and receptor- expressing cells isolated from one another, either with or without other surrounding cells. It revealed that both Notch1 and Notch2 could be *cis*-activated by the ligands Dll1 and Dll4. We also detected *cis*-activation across multiple cell types using both engineered and endogenous Notch components. Thus, *cis*-activation co-exists with *trans*-activation, and resembles it in many respects (*Figure 2*).

*Cis*-activation could potentially play functional roles in any Notch-dependent process. In neural stem cells (NSC), *cis*-activation appears to affect cell survival (*Figure 3*), thus suggesting that self-renewing cells can also be self-reliant, providing their own Notch signaling. This finding could help to explain how isolated stem cells can regenerate a complex tissue, as occurs in Lgr5+ intestinal crypt organoids (*Sato et al., 2009*) and mammary gland regeneration (*Stingl et al., 2006*), both of which are Notch-dependent.

Mathematical modeling suggests that *cis*-activation could broaden the capabilities of the Notch pathway: First, it could enable a 'negative' mode of *trans* Notch signaling, when high *cis*-activation is effectively inhibited by non-productive *trans* interactions (*Figure 5K*). This type of negative regulation is complementary to a previously described *trans*-inhibition mechanism where Notch1 activation by Dll4 was shown to be inhibited by the *trans*-ligand Jagged1 during angiogenesis (*Benedito et al., 2009*). Second, using a combination of *cis*-activation, *cis*-inhibition, and *trans*-activation could in principle enable a cell to discriminate the levels of its own (*cis*) ligands from those of its neighbors (*trans*) (*Figure 5I*). This property could be relevant for Notch-dependent fine-grained pattern formation through lateral inhibition circuits, in which cells coordinate their own Notch component levels with those of their neighbors (*Collier et al., 1996*; *Barad et al., 2010*; *Sprinzak et al., 2011*; *Formosa-Jordan and Ibañes, 2014b*).

In other systems, the combination of *cis* and *trans* signaling can produce interesting behaviors. For example, the EGFR ligand Heparin-binding EGF-like growth factor (HB-EGF) can exist in a membrane-anchored form that produces juxtacrine signaling or as a cleaved soluble form that can be involved in autocrine signaling. In MDCK cells, these isoforms produced distinct phenotypes, with cell survival and proliferation associated with the membrane-anchored isoform (*Raab and Klagsbrun, 1997*); *Singh et al., 2007*). Similarly, in yeast, rewiring of the mating pathway to create an autocrine signaling system revealed that qualitatively different behaviors ranging from quorum sensing to bimodality could be generated by tuning the relative strengths of *cis* and *trans* signaling (*Youk and Lim, 2014*). Looking ahead, it will be interesting to see how Notch *cis*-activation and *trans*-inhibition mechanisms combine in natural developmental contexts.

Mechanistically, it remains puzzling how *cis* interactions could lead to both activation and inhibition in a ligand concentration-dependent fashion. Productive *cis*-signaling and *trans*-signaling both appear to require ligand and receptor at the cell surface, suggesting *cis*-activation may involve a mechanism distinct from *cis*-interactions previously reported to occur within cellular endosomes (*Coumailleau et al., 2009*; *Fürthauer and González-Gaitán, 2009*). Along with our observations that *cis*- and *trans*-signaling also share a similar dependence on ligand-receptor affinity and ligand concentrations (*Figure 1—figure supplement 2C,D*), these results suggest that *cis*-activating complexes may resemble their *trans*-activating counterparts. Structural studies have shown that ligand-receptor binding could occur in both parallel and anti-parallel orientations (*Cordle et al., 2008*; *Luca et al., 2015*). It will therefore be interesting to see whether ligand-receptor complexes in different orientations can activate in a similar manner.

A striking feature of *cis*-activation is its non-monotonic dependence on ligand expression level, with *cis*-activation initially increasing as ligand levels increase, but ultimately giving way to *cis*-inhibition (*Figure 1*). Mathematical modeling enabled us to explore different ways in which *cis*-activation and *cis*-inhibition could coexist. The simplest model, which includes only a single type of *cis* ligand-receptor complex, could not reproduce this shift from *cis*-activation to *cis*-inhibition (*Figure 5A*, Model 1). By contrast, models that included distinct activating and inhibiting complexes generated these observed behaviors for a broad range of parameter values (*Figure 5A*, Model 2b-d). These results suggest that *cis*-activation and *cis*-inhibition involve the formation of distinct types of complexes.

C*is*-activation suggests new ways in which cells can integrate different types of Notch interactions. A more complete analysis of the *cis*- and *trans*- interactions among all ligand-receptor pairs, for different levels of Fringe expression, could provide a more comprehensive and predictive understanding of how cells with distinct component combinations signal to their neighbors, and to themselves, through Notch.

# Materials and methods

**Key resources table**

| Reagent type (species) or resource | Designation | Source or reference | Identifiers | Additional information |
|---|---|---|---|---|
| Gene (Mus musculus) | Dll1 | NCBI ID: 13388 | | |
| Gene (Homo sapiens) | Dll1 | NCBI ID: 28514 | | |
| Gene (Homo sapiens) | Dll4 | NCBI ID: 54567 | | |
| Gene (Mus musculus) | R-fringe | NCBI ID: 19719 | | |
| Gene (Homo sapiens) | Notch1 | NCBI ID: 4851 | | |
| Gene (Homo sapiens) | Notch2 | NCBI ID: 4853 | | |
| Cell line (Cricetulus griseus) | CHO-K1 | Thermo Fisher Scientific (T-REx CHO-K1) | Cat# R71807 RRID: CVCL_D586 | *Figures 1*, *2* and *4*; *Figure 1—figure supplements 1*, *2* and *3*; *Figure 2—figure supplements 1* and *2*; *Figure 4—figure supplement 1* |

*Continued on next page*

*Continued*

| Reagent type (species) or resource | Designation | Source or reference | Identifiers | Additional information |
|---|---|---|---|---|
| Cell line (Cricetulus griseus) | N1D1 | Derived from CHO-K1 | CHO-K1 expressing pEF-hNECD-Gal4esn + pcDNA5-TO-Dll1-T2A-H2B-mCherry + pEV-UAS-H2B-Citrine | *Figure 2* |
| Cell line (Cricetulus griseus) | N1D1 + Rfng | Derived from N1D1 | CHO-K1 expressing pEF-hNECD-Gal4esn + pcDNA5-TO-Dll1-T2A-H2B-mCherry + pEV-UAS-H2B-Citrine + pLenti-CMV-R-fringe-T2A-Puromycin | *Figures 1, 2* and *4; Figure 1—figure supplements 1* and *2; Figure 4—figure supplement 1* |
| Cell line (Cricetulus griseus) | N1WTD1 + Rfng | Derived from CHO-K1 | CHO-K1 expressing pcDNA3-hN1-mod1 + pcDNA5-TO-Dll1-mCherry + pEV-12xCSL-H2B-Citrine + piggyBac CMV-R-fringe+pCS-H2B-Cerulean | *Figure 1—figure supplement 3* |
| Cell line (Mus musculus) | NMuMG | ATCC | Cat# CRL-1636 (Wild-type cells used to transfect in piggyBac-12xCSL-H2B-Citrine) RRID:CVCL_0075 | *Figure 1—figure supplement 4* |
| Cell line (Mus musculus) | NMuMG + Dll1 | Derived from NMuMG | Base wild-type cell line expressing piggyBac-12xCSL-H2B-Citrine + piggyBac-TO-Dll1-T2A-H2B-mCherry-P2A-Hygromycin | *Figure 1—figure supplement 4* |
| Cell line (Mus musculus) | NMuMG N1D1 + Rfng | Derived from NMuMG | NMuMG ΔN2ΔJ1 expressing piggyBac-CMV-hNECD-Gal4-ANK-T2A-H2B-Cerulean + piggyBac CMV-TO Dll1-T2A-H2B-mCherry-P2A-Hygromycin + pEV-2xHS4-UAS-H2B-Citrine-T2A-tTS-2xHS4-Blast-T2A-rTetR-HDAC4-P2A-R-fringe | *Figure 1—figure supplement 3* |
| Cell line (Homo sapiens) | Caco-2 | ATCC (Caco-2 C2BBe1) | Cat# CRL-2102 (Wild-type cells used to transfect in pEV-12xCSL-H2B-Citrine) RRID:CVCL_1096 | *Figure 1—figure supplement 4* |

*Continued on next page*

*Continued*

| Reagent type (species) or resource | Designation | Source or reference | Identifiers | Additional information |
|---|---|---|---|---|
| Cell line (Cricetulus griseus) | N1D1 Pop | Derived from CHO-K1 | CHO-K1 with pEV-UAS-H2B-Citrine + pCS-H2B-Cerulean + piggyBac-TO-Dll1-T2A-H2B-mCherry + piggyBac-CMV-hN1ECD-Gal4 - Cell population (Pop) | *Figure 2* |
| Cell line (Cricetulus griseus) | N1D4 Pop | Derived from CHO-K1 | CHO-K1 with pEV-UAS-H2B-Citrine + pCS-H2B-Cerulean + piggyBac-TO-Dll4-T2A-H2B-mCherry + piggyBac-CMV-hN1ECD-Gal4 - Cell population (Pop) | *Figure 2*; *Figure 2— figure supplement 1* |
| Cell line (Cricetulus griseus) | N2D1 Pop | Derived from CHO-K1 | CHO-K1 with pEV-UAS-H2B-Citrine + pCS-H2B-Cerulean + piggyBac-TO-Dll1-T2A-H2B-mCherry + piggyBac-CMV-hN2ECD-Gal4 - Cell population (Pop) | *Figure 2*; *Figure 2— figure supplement 2* |
| Cell line (Cricetulus griseus) | N2D4 Pop | Derived from CHO-K1 | CHO-K1 with pEV-UAS-H2B-Citrine + pCS-H2B-Cerulean + piggyBac-TO-Dll4-T2A-H2B-mCherry + piggyBac-CMV-hN2ECD-Gal4 - Cell population (Pop) | *Figure 2*; *Figure 2— figure supplement 2* |
| Cell line (Mus musculus) | NSC | EMD Millipore | Cat# SCR029 (E14.5 mouse neural cortical stem cells - NSC) | *Figure 3*; *Figure 3—figure supplements 1* and 2 |
| Transfected construct (recombinant DNA) | pEV-UAS-H2B-Citrine | *Sprinzak et al., 2010* | N/A | Reporter for Notch1ECD-Gal4 receptor in CHO cells |
| Transfected construct (recombinant DNA) | pEV-2xHS4-UAS-H2B-Citrine-T2A-tTS-2xHS4-Blast-T2A-rTetR-HDAC4-P2A-R-fringe | This paper | N/A | Reporter for Notch1ECD-Gal4-ANK receptor in NMuMG cells (tTS was not relevant for this work and was inactivated by 4-epiTc); rTetR-HDAC4 was used to decrease Delta expression in the presence of Dox; Constitutive R-fringe expression in NMuMG cells |
| Transfected construct (recombinant DNA) | pEV-12xCSL-H2B-Citrine | *Sprinzak et al., 2010* | N/A | Reporter for Notch1 wild-type receptor in CHO and Caco-2 cells |

*Continued on next page*

*Continued*

| Reagent type (species) or resource | Designation | Source or reference | Identifiers | Additional information |
|---|---|---|---|---|
| Transfected construct (recombinant DNA) | pEF-hN1ECD -Gal4 | This paper | N/A | Notch1ECD-Gal4 synthetic receptor used in CHO clones |
| Transfected construct (recombinant DNA) | pX330 (CRISPR-Cas9 plasmid system) | *Cong et al., 2013* | N/A | Plasmid used to insert RNA guide sequence for CRISPR knockdown |
| Transfected construct (recombinant DNA) | piggyBac-12x CSL-H2B-Citrine | This paper | N/A | NotchWT reporter placed into NMuMG WT cells but not used in this study |
| Transfected construct (recombinant DNA) | piggyBac-CMV -hN1ECD-Gal4 | This paper | N/A | Notch1ECD-Gal4 synthetic receptor used in CHO populations |
| Transfected construct (recombinant DNA) | piggyBac-CMV -hN2ECD-Gal4 | This paper | N/A | Notch2ECD-Gal4 synthetic receptor used in CHO populations |
| Transfected construct (recombinant DNA) | pcDNA3- hN1-mod1 | *Sprinzak et al., 2010* | N/A | Wild-type Notch1 receptor used in CHO clones |
| Transfected construct (recombinant DNA) | piggyBac-CMV- hNECD-Gal4-ANK -T2A-H2B-Cerulean | This paper | N/A | Notch1ECD-Gal4-ANK synthetic receptor used in NMuMG clones |
| Transfected construct (recombinant DNA) | pcDNA5-TO-Dll1 -T2A-H2B-mCherry | *Nandagopal et al., 2018* | N/A | Inducible Delta-like1 ligand used in CHO clones and populations |
| Transfected construct (recombinant DNA) | pcDNA5-TO- Dll1- mCherry | *Sprinzak et al., 2010* | N/A | Inducible Delta-like1-mCherry fusion used in CHO clones with Notch1WT receptor |
| Transfected construct (recombinant DNA) | piggyBac-CMV-TO Dll1-T2A-H2B- mCherry-P2A- Hygromycin | *Nandagopal et al., 2018* | N/A | Inducible Delta-like1 ligand used in CHO populations and NMuMG clones |
| Transfected construct (recombinant DNA) | piggyBac-CMV-TO- Dll4-T2A-H2B-m Cherry-P2A- Hygromycin | *Nandagopal et al., 2018* | N/A | Inducible Delta-like4 ligand used in CHO populations |
| Transfected construct (recombinant DNA) | pLenti-CMV-R- fringe-T2A- Puromycin | This paper | N/A | Constitutive R-fringe expression in CHO N1D1 + Rfng cells |

*Continued on next page*

Continued

| Reagent type (species) or resource | Designation | Source or reference | Identifiers | Additional information |
|---|---|---|---|---|
| Transfected construct (recombinant DNA) | pCS-H2B-Cerulean | *Sprinzak et al., 2010* | N/A | Segmentation color used in CHO cells |
| Transfected construct (siRNA) | Allstar Negative Control | Qiagen | SI03650318 | Control siRNA, *Figure 3C* |
| Transfected construct (siRNA) | Dll1 siRNA | Thermo Fisher Scientific | Cat# 4390771 (ID: s65000) | Dll1 siRNA, *Figure 3C* |
| Antibody | rabbit anti-mouse Notch2 | Cell Signaling Technologies | Cat# 5732 RRID:AB_10693319 | WB (1:1000) |
| Antibody | rabbit anti-mouse Jagged1 | Cell Signaling Technologies | Cat# 2620 RRID:AB_659968 | WB (1:1000) |
| Antibody | rabbit anti-mouse GAPDH | Cell Signaling Technologies | Cat# 2118 RRID:AB_561053 | WB (1:3000) |
| Antibody | rabbit anti-mouse Dll1-ICD | Kindly provided by Gerry Weinmaster, UCLA | Antibody 88 c | WB (1:2000) |
| Antibody | ECL Rabbit IgG HRP-linked whole antibody from Donkey Secondary | GE Healthcare Life Sciences | Cat #NA934 RRID:AB_772206 | WB (1:2000) |
| Antibody | Anti-mouse AlexFluor 488 Secondary | Thermo Fisher Scientific | Cat# A21202 RRID:AB_141607 | ICC (1:1000) |
| Other | SuperSignal West Pico Chemiluminescent Substrate | Thermo Fisher Scientific | Cat# 34580 | as recommended per the manufacturer |
| Other | SuperSignal West Femto Chemiluminescent Substrate | Thermo Fisher Scientific | Cat# 34095 | as recommended per the manufacturer |
| Recombinant DNA reagent | Lipofectamine LTX plasmid transfection reagent | Thermo Fisher Scientific | Cat# 15338–100 | as recommended per the manufacturer |
| Recombinant DNA reagent | ViraPower Lentiviral Expression System | Thermo Fisher Scientific | Cat# K497500 | as recommended per the manufacturer |
| Peptide, recombinant protein | Recombinant mouse IgG2A Fc Protein | R and D Systems | Cat# 4460 MG-100 | 10 ug/ml |
| Peptide, recombinant protein | Recombinant mouse Dll1 Fc chimera | R and D Systems | Cat# 5026 DL-050 | 10 ug/ml |
| Peptide, recombinant protein | Recombinant mouse Notch-1 Fc chimera | R and D Systems | Cat# 5267-TK-050 | 10 ug/ml |
| Peptide, recombinant protein | Recombinant human Dll1ext-Fc fusion proteins | *Sprinzak et al., 2010* | Kindly provided by Irwin Bernstein, MD at Fred Hutchinson Cancer Research Center | 2.5 ug/ml |

*Continued on next page*

*Continued*

| Reagent type (species) or resource | Designation | Source or reference | Identifiers | Additional information |
|---|---|---|---|---|
| Chemical compound, drug | DAPT | Sigma Aldrich | Cat# D5942 | 1 uM (CHO cells); 10 uM (all other cells) |
| Chemical compound, drug | 4-epi tetracycline Hydrochloride | Sigma Aldrich | Cat# 37918 | 0–200 ng/ml |
| Chemical compound, drug | Doxycycline | Takara Bio USA Inc | Cat# 631311 | 1 ug/ml or 10 ug/ml |
| Chemical compound, drug | Dexamethasone | Sigma Aldrich | Cat# D4902 | 100 ng/ml |
| Commercial assay or kit | Miniprep kit | Qiagen | Cat# 27106 | |
| Commercial assay or kit | QIAquick PCR Purification kit | Qiagen | Cat# 28104 | |
| Commercial assay or kit | RNeasy mini kit for RNA extraction | Qiagen | Cat# 74106 | |
| Commercial assay or kit | iScript cDNA synthesis kit | Bio-Rad | Cat# 1708890 | |
| Commercial assay or kit | iQ SYBR Green Supermix | Bio-Rad | Cat# 1708880 | |
| Commercial assay or kit | SsoAdvanced Universal Probes Supermix | Bio-Rad | Cat# 172–5282 | |
| Commercial assay or kit | DNA HCR kit | Molecular Instruments | | |
| Software, algorithm | Cell segmentation and tracking | *Nandagopal et al., 2018* | N/A | *Figure 1*, *Figure 1—figure supplement 2* |
| Software, algorithm | FISH Transcript detection and quantification | This paper | https://github.com/nnandago/elife2018-dot_detection; *Nandagopal, 2018a* | MATLAB code for visualizing, segmenting, and detecting transcript dots in FISH-labeled cells. Used to generate *Figure 1—figure supplement 2*, *Figure 3—figure supplement 1* |
| Software, algorithm | cis-activation model | This paper | https://github.com/nnandago/elife2018-cis_activation_modeling; *Nandagopal, 2018b* | MATLAB code for modeling steady state concentrations of ligand, receptors and complexes for a range of parameters. Used to generate *Figure 5*, *Figure 5—figure supplement 5–* |

## Plasmids

The majority of constructs used in this study have been previously described (*Sprinzak et al., 2010*).
Briefly, the reporter for wild-type Notch activation was constructed from the 12xCSL plasmid (kind

gift from *Hansson et al., 2006*), while the UAS reporter for Notch1ECD-Gal4 receptor activation was a kind gift from S Fraser (*del Álamo et al., 2011*). The construct containing the full-length human wild-type Notch1 sequence was a kind gift from J Aster (*de Celis et al., 1997*). The Notch1-ECD-Gal4 plasmid was generated by replacing the Notch1ICD with amino acids 1–147 and 768–881 of the yeast Gal4 protein. This construct was further modified by incorporating the sequence of the ankyrin (ANK) domain from Notch1ICD (amino acids 1872–2144) 3' to the Gal4 sequence for use in the construction of the NMuMG cell lines. Design of the Notch2ECD-Gal4 plasmid was done in a manner similar to that of the Notch1ECD-Gal4 plasmid, but with incorporation of the expression cassette into a PiggyBac vector (System Biosciences, Palo Alto, CA) for efficient transfer to the cellular genome (Note: Notch1ECD-Gal4 was also incorporated into a PiggyBac vector when used in side-by-side comparisons with Notch2). The Notch ligand containing plasmids were based on the Tet-inducible system (Thermo Fisher Scientific, Waltham, MA). For the wild-type Notch1 cell line, we constructed a plasmid containing an inducible rat Dll1 coding sequence fused to the mCherry protein sequence. All other ligand plasmids were constructed containing an inducible human Dll1 or Dll4 coding sequence fused to a viral 2A sequence that allows for co-translation of a downstream H2B-mCherry protein sequence. In the NMuMG cells and in the CHO cell populations, the ligand plasmids were constructed within a PiggyBac vector. The Radical Fringe (Rfng) constructs used were based on those described by *LeBon et al., 2014*. For use in CHO-K1 cells, Rfng was cloned into a pLenti expression construct from the ViraPower Lentiviral Expression System (Thermo Fisher Scientific), modified with a CMV promoter and a puromycin resistance gene. For use with the NMuMG cells, Rfng was cloned into an insulated UAS reporter construct (UAS surrounded by 2 copies of the 2xHS4 insulating element) by adding a separate cassette containing the sequence for the blastomycin resistance gene followed by a viral 2A sequence connected to the sequence for the rTetR gene fused to a HDAC4-2A-Rfng sequence. rTetR-HDAC4 (rTetS) was used to suppress constitutive Notch ligand expression in the NMuMG cells with the addition of Doxycycline to the cell media. The pCS-H2B-Cerulean plasmid was described in (*Sprinzak et al., 2010*). All cloning was done using standard molecular biology cloning techniques.

## Cell culture and transfections
### CHO-K1
T-REx CHO-K1 cells from Thermo Fisher Scientific were cultured as described previously (*Sprinzak et al., 2010*; *LeBon et al., 2014*). Cells were tested by Thermo Fisher for the presence of Mycoplasma using the Gen−Probe Mycoplasma Tissue Culture NI (MTC−NI) Rapid Detection System and found to be negative. Transfection of CHO-K1 cells was performed in 24-well plates with 800–1000 ng of DNA using the Lipofectamine LTX plasmid transfection reagent (ThermoFisher Scientific). 24 hr post-transfection, cells were split into new 6-well plates and cultured for 1–2 weeks in media containing 400 ug/ml Zeocin, 600 ug/ml Geneticin, 300 ug/ml Hygromycin, 10 ug/ml Blasticidin, or 3 ug/ml Puromycin as appropriate, and surviving transfected cells were either used as polyclonal populations or subcloned by limiting dilution.

### NMuMG
NMuMG cells (ATCC, Manassas, VA) were cultured using the manufacturer's recommended culturing protocol with the addition of 1 mM Sodium Pyruvate and 100 U ml-1 penicillin, 100 μg ml-1 streptomycin (Thermo Fisher Scientific) to the media. These cells were authenticated by ATCC and found to be mycoplasma free by Hoechst DNA stain and agar culture. Transfection, selection and clonal isolation of NMuMG cells were performed similarly to CHO-K1 cells.

### Caco-2
Caco-2 C2BBe1 cells from ATCC were cultured in Dulbecco's modified Eagle's medium (DMEM, Thermo Fisher Scientific) supplemented with 10% Tet-System Approved Fetal Bovine Serum (Takara Bio USA Inc, Mountain View, CA), 2 mM L-Glutamine, 100 U ml-1 penicillin, 100 μg ml-1 streptomycin, 1 mM sodium pyruvate, and 1X MEM Non-Essential Amino Acids Solution (Thermo Fisher Scientific). ATCC uses morphology, karyotyping, and PCR based approaches to confirm the identity of human cell lines and to rule out both intra- and interspecies contamination. These include an assay to detect species specific variants of the cytochrome C oxidase I gene (COI analysis) to rule out

inter-species contamination and short tandem repeat (STR) profiling to distinguish between individual human cell lines and rule out intra-species contamination. Transfections were performed by following the Thermo Fisher Lipofectamine LTX protocol which was optimized for Caco-2 cells. 24 hr after transfection, Caco-2 cell populations were plated for experiments.

## Neural Stem Cells

Neural stem cells derived from the E14.5 mouse cortex were purchased from EMD Millipore (Burlington, MA, Catalog No. SCR029) and cultured according to the manufacturer's protocols. Briefly, tissue-culture surfaces were coated overnight with poly-L-ornithine (10 ug/ml, Sigma-Aldrich Catalog No. P3655) and Laminin (Sigma-Aldrich Catalog No. L2020). For standard culture, cells were then plated in Neurobasal medium (EMD Millipore, Catalog No. SCM033) in the presence of 20 ng/ml recombinant FGF (EMD Millipore Catalog No. GF003), 20 ng/ml EGF (Millipore Catalog No. GF001), and Heparin (Sigma-Aldrich Catalog No. H3149). Cells were passaged using ESGRO Complete Accutase (Millipore Catalog No. SF006), cryo-preserved in medium + 10% DMSO, and typically used for experiments within six passages.

## Lentiviral production and infection

Lentivirus was produced using the ViraPower Lentiviral Expression System (Thermo Fisher Scientific). Briefly, 293FT producer cells were transfected with our pLenti expression construct along with the packaging plasmid mix. 48 hr post-transfection, virus containing cell media was collected, centrifuged to remove cell debris and filtered through a 0.45 um filter (EMD Millipore). Viral supernatant was added 1:2 to sparsely plated CHO-K1 cells in a total volume of 400 ul media in a 24-well plate and incubated at 37°C, 5% $CO_2$. 24 hr post-infection, virus containing media was removed, and cells were plated under limiting dilution conditions in 96-well plates for clonal selection. Expression of the integrated gene was checked by qRT-PCR analysis.

## RNASeq

RNA was isolated from cells using the RNeasy Kit (Qiagen, Hilden, Germany) and cDNA libraries were prepared according to standard Illumina protocols at the Millard and Muriel Jacobs Genetics and Genomics Laboratory at Caltech. SR50 sequencing (10 libraries/lane) with a sequencing depth of 20–30 million reads was performed on a HiSeq2500. Reads were assembled, aligned and mapped to the mouse genome (mm10 assembly) using Tophat2 (*Kim et al., 2013*) or RNAstar (*Dobin et al., 2013*) on the Galaxy platform (https://usegalaxy.org). Cufflinks was subsequently used on mapped reads to calculate FPKM values. RNAseq data for gene expression in neural stem cells cultured in reduced or standard growth factor conditions, and treated with or without DAPT, is available on GEO (accession: GSE113937).

## CRISPR-Cas9 knockout of endogenous NMuMG Notch2 and Jagged1

Endogenous Notch2 and Jagged1 genes were knocked out in NMuMG cells using the CRISPR-Cas9 plasmid system developed by the Zhang Lab at MIT (*Cong et al., 2013*). Cloning was done according to the published protocol using the pX330 plasmid and inserting a guide sequence using the following oligos for targeting mouse Notch2 or Jagged1:

## Notch2

mN2 C2 OligoF: 5'-CACCGGGTGGTACTTGTGTGCCGCA-3'
mN2 C2 OligoR: 5'-AAACTGCGGCACACAAGTACCACCC-3'

## Jagged1

mJ1 C1 OligoF: 5'-CACCGCGGGTGCACTTGCGGTCGCC-3'
mJ1 C1 OligoR: 5'-AAACGGCGACCGCAAGTGCACCCGC-3'

The guide sequence modified pX330 plasmids were transfected into NMuMG cells using the standard Lipofectamine LTX protocol (Thermo Fisher Scientific). 48 hr post-transfection, genomic DNA was harvested from the cell population, and guide sequence function was analyzed using the SURVEYOR Mutation Detection Kit (Integrated DNA Technologies Inc, Skokie, IL). After genomic knockout mutation was verified, transfected cells were placed under clonal selection using limiting

dilution. Genomic DNA was isolated from clones and used to PCR amplify targeted sequences using the following primers (Integrated DNA Technologies Inc):

## Notch2
mN2 C2F: 5'-GTCACCCGTCTGGTATTTTGTTAC-3'
mN2 C2R: 5'-GAGCTGCTGTGATCGAAGTG-3'

## Jagged1
mJ1 C1F: 5'-CCAAAGCCTCTCAACTTAGTGC-3'
mJ1 C1R: 5'-CTTAGTTTTCCCGCACTTGTGTTT-3'

PCR products were purified using the QIAquick PCR Purification Kit (Qiagen) and sent for sequencing (Laragen Inc, Culver City, CA) to determine clones that contained gene knockout mutations. Gene knockout was also verified in NMuMG clones by western blot analysis of endogenous Notch2 and Jagged1 protein. Briefly, protein was harvested by lysing cells with Cell Lysis Buffer (Cell Signaling Technologies, Danvers, MA) supplemented with NuPAGE LDS Sample Buffer (Thermo Fisher Scientific), 80 mM DTT (Sigma-Aldrich, St. Louis, MO), 1.2 mM PMSF (Cell Signaling Technologies), and Halt Proteinase Inhibitor (Thermo Fisher Scientific). Protein lysate was heated at 95$^\circ$C, followed by incubation on ice for 2 min. Lysate was centrifuged at 13 K rpm for 10 min. at 4$^\circ$C, and 1/5$^{th}$ was loaded onto a 4–12% Bis-Tris gel (Thermo Fisher Scientific). After soaking the gel in 20% ethanol for 10 min., protein was transferred from the gel to a nitrocellulose membrane using the iBlot Dry Blotting System (Thermo Fisher Scientific). The blot was blocked in 1xTBST, 5% dry milk, 2% BSA for 1 hr at room temp followed by overnight incubation at 4$^\circ$C with either a rabbit anti-mouse Notch2 antibody at 1:1000 (Cell Signaling Technologies, #5732) or a rabbit anti-mouse Jagged1 antibody at 1:1000 (Cell Signaling Technologies, #2620) together with a rabbit anti-mouse GAPDH loading control antibody at 1:3000 (Cell Signaling Technologies, #2118). The next day, the blot was washed and incubated with an anti-rabbit HRP conjugated antibody at 1:2000 (GE Healthcare Life Sciences, Marlborough, MA) for 1 hr at room temp followed by washing and band detection using SuperSignal West Pico Chemiluminescent Substrate (Thermo Fisher Scientific).

## Availability assay for Notch1ECD-Gal4 or Notch1ECD-Gal4-ANK in NMuMG cells

Surface staining of either Notch1ECD-Gal4 or Notch1ECD-Gal4-ANK was performed using the availability assay as described previously (*LeBon et al., 2014*). Briefly, cells were washed in phosphate buffered saline (PBS) and blocked in a PBS solution containing 2% BSA and 100 ug/ml CaCl$_2$ while rocking for 40 min at room temperature. After blocking, cells were rocked in a PBS solution containing 2% FBS, 100 ug/ml CaCl$_2$, and 10 ug/ml of recombinant mouse Dll1-Fc (rmDll1-Fc) for 1 hr at room temperature. The recombinant Dll1 protein can bind to the available (free) Notch receptors at the cell surface. After the 1 hr incubation, cells were washed 3x with PBS and incubated with a secondary antibody conjugated to an Alexa Fluor 488 dye. Cells were rocked at room temperature in the dark for 1 hr, washed 3x with PBS, and Notch localization on the cell surface was imaged on an EVOS FL Auto Cell Imaging system (Thermo Fisher Scientific).

## qRT-PCR

Expression of Radical Fringe (Rfng) in clones was determined by quantitative RT-PCR. RNA was isolated from clonal cells using the RNeasy Mini Kit (Qiagen) following the manufacturer's protocol. 200–500 ng RNA was used to make cDNA using the iScript cDNA Synthesis Kit (Bio-Rad Laboratories, Hercules, CA). 2 ul of cDNA was used in the qPCR reaction along with SsoAdvanced Universal Probes Supermix (Bio-Rad Laboratories), and primers/probes (Integrated DNA Technologies Inc) as follows:

Rfng
Probe: 5'−6-FAM/ZEN/3' IBFQ-CTCGTGAGATCCAGGTACGCAGC-3'
Primer 1: 5'-TCATTGCAGTCAAGACCACTC-3'
Primer 2: 5'-CGGTGAAAATGAACGTCTGC-3'
b-Actin (Housekeeping Gene for CHO-K1 cells)
Probe: 5'-HEX/ZEN/3' IBFQ-ACCACACCTTCTACAACGAGCTGC-3'

Primer1: 5'-ACTGGGACGATATGGAGAAG-3'
Primer2: 5'-GGTCATCTTTTCACGGTTGG-3'
GAPDH (Housekeeping Gene for NMuMG cells)
Probe: 5'-HEX/ZEN/3' IBFQ-AGGAGCGAGACCCCACTAACATCA-3'
Primer1: 5'-CTCCACGACATACTCAGCAC-3'
Primer2: 5'-CCACTCACGGCAAATTCAAC-3'

Samples were run in triplicate on a CFX96 Touch Real-Time PCR Detection System (Bio-Rad Laboratories), and relative gene expression levels were calculated using the standard delta-delta Cq method.

Expression of Hes1 in wild-type and Dll1 transfected NMuMG cells was determined by quantitative RT-PCR as described above with the use of IQ SYBR Green Supermix (Bio-Rad Laboratories). The 'mouse Hes1 primer set 2' primers (Integrated DNA Technologies Inc) from *Nandagopal et al., 2018* were used in the qPCR reaction.

## Plate-bound Dll1

Coating of tissue culture plates with recombinant human Dll1$^{ext}$-Fc fusion proteins (kind gift from I Bernstein) was done as previously reported (*Nandagopal et al., 2018*). Briefly, Dll1$^{ext}$-Fc proteins were diluted to 2.5 ug/ml in 1xPBS (Thermo Fisher Scientific), and the solution was used to coat the tissue-culture surface for 1 hr at room temperature with rocking. Post-incubation, the solution was removed and cells were plated for the experiment.

## Single molecule HCR-FISH detection of Notch targets in NMuMG and NSCs

### Experimental protocols

NMuMG cells - Cells were pre-treated with 10 uM DAPT for at least 2 days before plating at $3 \times 10^3$ cells/24-well tissue culture treated ultrathin glass film bottom plates (Eppendorf, Hamburg, Germany) with or without DAPT. All cells were treated with 100 ng/ml Dexamethasone (Sigma-Aldrich) at the time of plating. 6 hr post-DAPT removal, cells were fixed using 4% formaldehyde.

Neural Stem cells - NSCs were cultured in standard high growth factor containing media (see cell culture and transfections section above), and plated on 10 ug/ml poly-L-ornithine and 50 ug/ml Laminin-coated glass plates, at a surface density of ~4 cells/mm$^2$. At the time of plating, cells were transferred to low growth factor conditions (0.1 ng/ml EGF, 5 ug/ml Heparin, no FGF), with or without 10 µM DAPT. 6 hr post plating, cells were fixed using 4% formaldehyde.

Prescribed protocols were followed for hybridizing DNA probes to target genes (*Hes1*, *Hey1*, and *Hes5*) and amplifying bound probes (*Choi et al., 2018*). Briefly, fixed cells were incubated overnight at 37°C with 10 pairs of probes per gene diluted in 30% formamide-containing buffer. Subsequently, probes were removed and cells were washed at 37°C. Then, DNA amplifiers, designed to detect bound probes and coupled to one of three Alexa Fluor dyes (488, 546, or 647), were added to the sample, and amplification allowed to proceed at room temperature for ~50 min. Samples were then washed in high salt solution (5x SSC with Tween), and stained with DAPI, prior to imaging.

### Imaging

Samples were imaged at 60x (1.3 NA, oil) on an inverted epi-fluorescence microscope (Nikon Ti: Nikon Instruments Inc, Melville, NY) equipped with an LED light source (Lumencor, Beaverton, OR) and hardware autofocus. Fields of view that contained between 1–3 well-separated cells were picked manually and Z-stacks were acquired over 16 µm at each position.

### Analysis

Custom MATLAB (2015a, Mathworks) software was used to semi-automatically segment cells based on autofluorescence in the 488 channel. mRNA transcripts typically appeared as 3–5 voxel-wide high-intensity dots in the images. Previously used MATLAB software for detecting dots (*Frieda et al., 2017*) was adapted to automatically detect dots in images based on user-defined thresholds. For direct comparison, the same thresholds were applied to data from DAPT-treated and untreated samples. Segmentation and dot detection code have been deposited on GitHub

### *Cis*-activation and relative density assays

CHO-K1 engineered cell lines were pre-incubated with 1 uM of the gamma-secretase inhibitor DAPT (Sigma-Aldrich) and various concentrations of the tetracycline analog, 4-epiTetracycline (4-epiTc, Sigma-Aldrich) 48 hr prior to the setup of assays. For the *cis*-activation assay, cells were washed to remove DAPT, counted, and plated sparsely at 5 K cells per 24-well plate, surrounded by 150K wild-type CHO-K1 cells. 4-epiTc was added back into the media (0, 20, 35, 50, 80, or 200 ng/ml) and the cells were either incubated at 37°C, 5% $CO_2$ for <24 hr before analysis by flow cytometry or imaged using time-lapse microscopy. For the 'control' *cis*-activation assay, 4K 4-epiTc pre-induced N1D1 + Rfng cells were plated along with 4K N1 receiver cells (no ligand present), and 750K CHO-K1 cells per 6-well plate. Cells were analyzed for activation by flow cytometry (see 'Flow cytometry analysis' section below). Relative density assays were performed using the same setup conditions as the *cis*-activation assay, but with varying ratios of engineered:wild-type cells plated. Keeping total cell numbers at 150 K cells per 24-well plate, either 5K, 10K, 25K, 50K, 75K or 100K engineered cells were plated along with wild-type CHO-K1 cells. For the *cis*-activation and relative density assays using Blebbistatin treated cells, assay setup was done exactly as mentioned above but with the addition of 10 uM (±)-Blebbistatin (Sigma-Aldrich) added to the cell media at the time of plating.

For NMuMG cells, *cis*-activation and density assays were performed just like those with CHO-K1 cells. However, cells were pre-incubated in 10 uM DAPT with or without the addition of 1 ug/ml or 10 ug/ml Doxycycline (Takara Bio USA Inc) for 3 days in order to decrease ligand expression levels prior to assay setup. Cells were treated with 100 ng/ml Dexamethasone (Sigma-Aldrich) at time of plating for each assay.

Caco-2 cells were pre-incubated with 10 uM DAPT for 1 day prior to transfection. 24 hr post-12xCSL reporter transfection, the cells were washed, counted and plated sparsely at 3.5 K cells in a 24-well plate for the *cis*-activation assay, with or without the addition of DAPT. < 24 hr after plating, cells were analyzed by flow cytometry.

### *Cis*-activation assay in suspension

For performing the *cis*-activation assay with cells in suspension, 24-well 10 mm diameter glass No. 1.5 coverslip plates (MatTek Corp., Ashland, MA) were coated with the siliconizing reagent Sigma-cote (Sigma-Aldrich) to prevent cells from adhering to the plate surface. Cells were plated as mentioned previously and placed on a rocker at 37°C, 5% $CO_2$ overnight before analysis by flow cytometry.

### Notch receptor/ligand blocking assay

Engineered CHO-K1 cells were pre-incubated in 1 uM DAPT, with and without the addition of 4-epiTc, for 2 days prior to the start of the assay. Cells were then incubated with either 10 ug/ml mouse $IgG_{2A}$ control protein or 10 ug/ml mouse Notch1 Fc chimera protein (R and D Systems, Minneapolis, MN) along with DAPT and 4-epiTc overnight at 37°C, 5% $CO_2$. The next day, cells were washed, counted, and plated at 5 K cells per 24-well plate along with 150K wild-type CHO-K1 cells for a *cis*-activation assay, or at 150 K cells per 24-well plate for a relative density assay with the addition of 4-epiTc. Cells grown similarly, but in the absence of 4-epiTc, were used as a control. < 24 hr post-plating, cells were analyzed by flow cytometry.

### NSC survival assay
Experimental Protocols

NSCs, cultured in standard high growth factor containing media (see cell culture and transfections section above), were plated on 10 ug/ml poly-L-ornithine and 5 ug/ml Laminin-coated plastic surfaces (12-well TC-treated plates, Corning Inc) at a surface density of ~20 cells/mm$^2$. At the time of plating, cells were transferred to low growth factor conditions (0.1 ng/ml EGF, 5 ug/ml Heparin, no FGF), with or without 10 μM DAPT. 12 hr post plating, cells were fixed using 4% formaldehyde. Samples were then stained with DAPI prior to imaging.

## Imaging

Samples were imaged at 20x (0.75 NA, air) in an inverted epi-fluorescence microscope (Olympus IX81) equipped with an LED lightsource (XCite LED) and hardware ZDC2 autofocus. 484 600 μm x 600 μm fields of view were acquired from across the well for each sample.

## Analysis

Custom MATLAB (2015a, Mathworks) software was used to automatically segment nuclei based on DAPI staining. The number of nuclei were counted for each of the different samples.

## Dll1 knockdown in NSCs

### Experimental Protocols

$1 \times 10^6$ NSCs, cultured in standard high growth factor containing media (see cell culture and transfections section above), were nucleofected with 100 pmol of Silencer Select siRNA targeting mouse Dll1 (5'-CGAUGACCUCGCAACAGAAtt-3', Life Technologies) or Allstar negative control siRNA (Qiagen). Cells were nucleofected using mouse neural stem cell-specific reagents (Lonza Bioscience) in an Amaxa 2b nucleofector (Program A-033, Lonza Bioscience). They were subsequently cultured for 48 hr and then plated sparsely, in duplicate, for a cell survival assay as described above.

### Imaging

Samples were imaged at 20x (0.75 NA, air) in an inverted epi-fluorescence microscope (Nikon Ti) equipped with an LED lightsource (Lumencor Sola). 49 fields of view were acquired from across the well for each sample.

### Analysis

Custom MATLAB (2015a, Mathworks) software was used to automatically segment nuclei based on DAPI staining. The number of nuclei were then counted for each of the different samples.

## Western blot analysis of Dll1 knockdown in NSCs

NSCs treated with control or Dll1-targeting siRNA and plated for a cell survival assay, as mentioned in the previous section, were collected for western blot analysis of Dll1 knockdown. More specifically, an equal number of siRNA treated cells were harvested for protein and processed as previously described (see CRISPR-Cas9 knockout of endogenous NMuMG Notch2 and Jagged1 section). The blot was blocked in 1xTBST, 5% dry milk, 2% BSA for 1 hr at room temp followed by overnight incubation at 4°C with a rabbit anti-Dll1-ICD antibody at 1:2000 (88c, a generous gift from Gerry Weinmaster at UCLA) together with a rabbit anti-mouse GAPDH loading control antibody at 1:3000 (Cell Signaling Technologies, #2118). The next day, the blot was washed and incubated with an anti-rabbit HRP conjugated antibody at 1:2000 (GE Healthcare Life Sciences) for 1 hr at room temp followed by washing and band detection using SuperSignal West Femto Chemiluminescent Substrate (Thermo Fisher Scientific).

## Time-lapse setup, image acquisition and analysis

### Experimental setup

For imaging, CHO-K1 cells were plated in 24-well 10 mm diameter glass No. 1.5 coverslip plates (MatTek Corp.) coated with 5 ug/ml hamster Fibronectin (Oxford Biomedical Research, Rochester Hills, MI) in complete cell media. NMuMG cells were plated in 24-well tissue culture treated ultrathin glass film bottom plates (Eppendorf) in complete cell media.

### Acquisition

Movies were acquired at 20X (0.75 NA) on an Olympus IX81 inverted epifluorescence microscope (Olympus, Tokyo, Japan) equipped with hardware autofocus (ZDC2), an iKon-M CCD camera (Andor, Concord, MA) and an environmental chamber maintaining cells at 37°C, 5% $CO_2$ with humidity throughout the length of the movie. Automated acquisition software (Metamorph, Molecular Devices, San Jose, CA) was used to acquire images every 30 min in multiple channels (YFP, RFP, CFP) or differential interference contrast (DIC), from multiple stage positions.

## Analysis

Custom MATLAB code (2013a, MathWorks) was used to segment cell nuclei in images based on constitutive Cerulean fluorescence. Briefly, the segmentation procedure uses built-in edge detection MATLAB functions and adaptive thresholds to detect nuclear segments. Nuclear segments were then matched in pairs of images corresponding to consecutive time frames, and thus tracked through the duration of the movie. Single-cell tracks were subsequently curated manually to correct for errors in segmentation/tracking. Fluorescence data was extracted from nuclear segments by calculating the integrated fluorescence within the segment and subtracting a background fluorescence level estimated from the local neighborhood of the segment. This fluorescence was linearly interpolated across time frames where nuclei could not be segmented automatically. Division events were detected automatically, and fluorescence traces were corrected for cell division by adding back fluorescence lost to sister cells. The resulting 'continuized' traces were smoothed and the difference in fluorescence between consecutive time frames was calculated. A smoothed version of this difference was used as an estimate of production rate of the fluorescent protein.

## Flow cytometry analysis

For analysis of cells by flow cytometry, cells were trypsinized in 0.25% Trypsin-EDTA (Thermo Fisher Scientific) and resuspended in 1x Hanks Balanced Salt Solution (Thermo Fisher Scientific) supplemented with 2.5 mg/ml Bovine Serum Albumin (Sigma-Aldrich). Resuspended cells were filtered using 40 um cell strainers (Corning Inc, Corning, NY) into U-bottom 96-well tissue culture treated plates. Cells were analyzed on a MACSQuant VYB Flow Cytometer (Miltenyi Biotech, Bergisch Gladbach, Germany) located at the Caltech Flow Cytometry Facility (Caltech, Pasadena, CA). Data was analyzed in MATLAB using custom software (EasyFlow) (*Antebi et al., 2017*), and forward and side-scatter profiles were used to gate on the proper cell populations. Fluorescence intensity of single-cells was measured for each appropriate channel.

## Mathematical models

### Note

MATLAB Code for models has been deposited on GitHub (https://github.com/nnandago/elife2018-cis_activation_modeling; copy archived at https://github.com/elifesciences-publications/elife2018-cis_activation_modeling).

### Models

The models analyzed here attempt to recapitulate the behavior of the system at steady state. Components of the system include free Notch receptor (N) and free Delta ligand (D) and, depending on the model, *cis*- and *trans*-complexes ($C^+$/$C^-$, and T, respectively) between ligands and receptors. In all models, N and D are produced at a rate of $\alpha_N$ and $\alpha_D$, and degraded at the rate of $\gamma_N$ and $\gamma_D$, respectively.

### I) Model 0

This model assumes that N and D interact at a rate $k^+$ to produce a single type of *cis*-complex, $C^+$, which dissociates at a rate $k^-$, and is degraded at a rate $\gamma_{C+}$. That is,

$$\frac{dN}{dt} = \alpha_N - \gamma_N N - k^+_{C+} ND + k^-_{C+} C^+$$

$$\frac{dD}{dt} = \alpha_D - \gamma_D D - k^+_{C+} ND + k^-_{C+} C^+$$

$$\frac{dC^+}{dt} = -\gamma_{C+} C^+ + k^+_{C+} ND - k^-_{C+} C^+$$

### II) Model 1

This model assumes that N and D can interact to produce two distinct *cis*-complexes, active $C^+$ and inactive $C^-$. These complexes are formed through similar second-order interactions between N and D, occurring with different rate coefficients. They similarly dissociate and degrade at different rates, and cannot interconvert.

$$\frac{dN}{dt} = \alpha_N - \gamma_N N - k_{C+}^+ ND + k_{C+}^- C^+ - k_{C-}^+ ND + k_{C-}^- C^-$$

$$\frac{dD}{dt} = \alpha_D - \gamma_D D - k_{C+}^+ ND + k_{C+}^- C^+ - k_{C-}^+ ND + k_{C-}^- C^-$$

$$\frac{dC^+}{dt} = -\gamma_{C+} C^+ + k_{C+}^+ ND - k_{C+}^- C^+$$

$$\frac{dC^-}{dt} = -\gamma_{C-} C^- + k_{C-}^+ ND - k_{C-}^- C^-$$

## III) Models 2a-d

These models also assume that N and D can interact to produce two distinct *cis*-complexes, $C^+$ and $C^-$. However, in these models, $C^-$ requires $C^+$ for its formation (a-c) or the stoichiometry of inactive $C^-$ formation is higher than that of $C^+$ (d).

**a)** $N + D \leftrightarrow C^+, \ \ N + D + C^+ \leftrightarrow C^-$

$$\frac{dN}{dt} = \alpha_N - \gamma_N N - k_{C+}^+ ND + k_{C+}^- C^+ - k_{C-}^+ NDC^+ + k_{C-}^- C^-$$

$$\frac{dD}{dt} = \alpha_D - \gamma_D D - k_{C+}^+ ND + k_{C+}^- C^+ - k_{C-}^+ NDC^+ + k_{C-}^- C^-$$

$$\frac{dC^+}{dt} = -\gamma_{C+} C^+ + k_{C+}^+ ND - k_{C+}^- C^+ - k_{C-}^+ NDC^+ + k_{C-}^- C^-$$

$$\frac{dC^-}{dt} = -\gamma_{C-} C^- + k_{C-}^+ NDC^+ - k_{C-}^- C^-$$

**b)** $N + D \leftrightarrow C^+, \ \ C^+ + C^+ \leftrightarrow C^-$

$$\frac{dN}{dt} = \alpha_N - \gamma_N N - k_{C+}^+ ND + k_{C+}^- C^+$$

$$\frac{dD}{dt} = \alpha_D - \gamma_D D - k_{C+}^+ ND + k_{C+}^- C^+$$

$$\frac{dC^+}{dt} = -\gamma_{C+} C^+ + k_{C+}^+ ND - k_{C+}^- C^+ - k_{C-}^+ (C^+)^2 + 2k_{C-}^- C^-$$

$$\frac{dC^-}{dt} = -\gamma_{C-} C^- + k_{C-}^+ (C^+)^2 - k_{C-}^- C^-$$

**c)** $N + D \leftrightarrow C^+, \ \ D + C^+ \leftrightarrow C^-$

$$\frac{dN}{dt} = \alpha_N - \gamma_N N - k_{C+}^+ ND + k_{C+}^- C^+$$

$$\frac{dD}{dt} = \alpha_D - \gamma_D D - k_{C+}^+ ND + k_{C+}^- C^+ - k_{C-}^+ DC^+ + k_{C-}^- C^-$$

$$\frac{dC^+}{dt} = -\gamma_{C+} C^+ + k_{C+}^+ ND - k_{C+}^- C^+ - k_{C-}^+ DC^+ + k_{C-}^- C^-$$

$$\frac{dC^-}{dt} = -\gamma_{C-} C^- + k_{C-}^+ DC^+ - k_{C-}^- C^-$$

**a)** $N + D \leftrightarrow C^+, \ \ 2N + 2D \leftrightarrow C^-$

$$\frac{dN}{dt} = \alpha_N - \gamma_N N - k_{C+}^+ ND + k_{C+}^- C^+ - k_{C-}^+ N^2 D^2 + 2k_{C-}^- C^-$$

$$\frac{dD}{dt} = \alpha_D - \gamma_D D - k_{C+}^+ ND + k_{C+}^- C^+ - k_{C-}^+ N^2 D^2 + 2k_{C-}^- C^-$$

$$\frac{dC^+}{dt} = -\gamma_{C+} C^+ + k_{C+}^+ ND - k_{C+}^- C^+$$

$$\frac{dC^-}{dt} = -\gamma_{C-} C^- + k_{C-}^+ N^2 D^2 - k_{C-}^- C^-$$

## IV) Model 2c + trans-interactions

To model the effect of trans-interactions in the context of model 2c, it was assumed that the *trans*-complex T is formed through interactions between N and *trans* Delta ($D_{trans}$, assumed to be constant). The rate coefficients of its formation, dissociation, and degradation are assumed to be the same as that of $C^+$.

$$N + D_{cis} \leftrightarrow C^+, \quad D_{cis} + C^+ \leftrightarrow C^-, \quad N + D_{trans} \leftrightarrow T$$

$$\frac{dN}{dt} = \alpha_N - \gamma_N N - k_{C+}^+ N D_{cis} + k_{C+}^- C^+ - k_{C+}^+ N D_{trans} + k_{C+}^- T$$

$$\frac{dD_{cis}}{dt} = \alpha_D - \gamma_D D_{cis} - k_{C+}^+ N D_{cis} + k_{C+}^- C^+ - k_{C-}^+ D_{cis} C^+ + k_{C---}^- C^-$$

$$\frac{dC^+}{dt} = -\gamma_{C+} C^+ + k_{C+}^+ N D_{cis} - k_{C+}^- C^+ - k_{C-}^+ D_{cis} C^+ + k_{C---}^- C^-$$

$$\frac{dT}{dt} = -\gamma_{C+} T + k_{C+}^+ N D_{trans} - k_{C+}^- T$$

## Parameter scan, numerical simulations, and analyses

Model 0 contains 6 parameters ($\alpha_N$, $\gamma_N$, $\gamma_D$, $\gamma_{C+}$, $k_{C+}^+$, $k_{C+}^-$), while Models 1 and 2 contain 3 additional parameters ($\gamma_{C-}$, $k_{C-}^+$, $k_{C-}^-$). Using the built-in *lhsdesign* function in MATLAB (2015a, Mathworks), the Latin Hypercube Sampling algorithm was applied to pick 10,000 parameters, each in the range $10^{-2}$ to $10^2$. For each parameter set, the model was simulated for each of 10 values of $\alpha_D$, logarithmically spanning a 100-fold range around the sampled value of $\alpha_N$. The *fsolve* function, with initial conditions $[\frac{\alpha_N}{\gamma_N}, \frac{\alpha_D}{\gamma_D}, 1, 1]$ for N, D, C$^+$, C$^-$, respectively, was used to numerically approximate the steady state solution for each parameter set.

For each solution, the following features of the $\alpha_D$ vs. C$^+$ profile were calculated: the relative value of $\alpha_D$ at which C$^+$ was maximum ('C-max'), and the fractional increases in C-max relative to its value at the lowest and highest values of $\alpha_D$. Parameters that produced C$^+$ profiles that peaked between the 1st and 8th value of $\alpha_D$ were deemed to be non-monotonic.

For the *trans*-interaction model, first the values of D$_{cis}$ obtained at D$_{trans}$ = 0 were calculated for each parameter set. For subsequent simulations, the values of D$_{trans}$ were chosen to be the same as that of these D$_{cis}$ values, that is D$_{cis}$ produced in the absence of *trans*-ligand.

## Statistics

No statistical method was used to determine sample sizes. The sample sizes used were based on general standards accepted by the field. The number of replicates used for each experimental analysis is listed in the figure legends. All replicates are biological replicates, corresponding to measurements performed on distinct biological samples, as opposed to performing the same tests multiple times on a single sample (technical replicates). *P*-values for *Figures 1* and *3* were calculated using the two-sided KS-test. All pairwise comparisons between samples fulfilled the criterion n1*n2/ (n1 + n2)$\geq$4, where n1 and n2 represent the number of data points in two samples. Under this condition, the KS-statistic is greater than twice the inverse of the Kolmogorov statistic, and the calculated *P*-value is accurate. The non-parametric nature of the KS-test obviates the need to make assumptions regarding the shape of the distributions being compared. *P*-values for Caco-2 cell measurements (*Figure 1—figure supplement 4*) and CHO cell surface binding assay measurements (*Figure 4*) were calculated using the one-sided Student T-test, which assumes that random error in the measurement follows a normal (Gaussian) distribution.

## Acknowledgements

This work was supported by Howard Hughes Medical Institute (MBE), and the Defense Advanced Research Projects Agency under Contract No. HR0011-16-0138, by the National Institutes of Health grant R01 HD075335 and the NSF under grant EFRI 1137269. NN was a Howard Hughes Medical Institute International Student Research fellow. We thank Pulin Li, Mark Budde, Heidi Klumpe, Ronghui Zhu, Rachael Kuintzle, Laurent Potvin-Trottier and James Linton for critical feedback on the manuscript. Harry Choi and Colby Calvert, Caltech Flow Cytometry Facility, Caltech Biological Imaging Facility, and the Millard and Muriel Jacobs Genetics and Genomics Laboratory at Caltech provided essential technical assistance.

## Additional information

### Funding

| Funder | Grant reference number | Author |
|---|---|---|
| National Institutes of Health | R01 HD075335 | Nagarajan Nandagopal<br>Leah A Santat<br>Michael B Elowitz |
| Howard Hughes Medical Institute | MBE | Nagarajan Nandagopal<br>Leah A Santat<br>Michael B Elowitz |
| Defense Sciences Office, DARPA | HR0011-16-0138 | Nagarajan Nandagopal |
| National Science Foundation | EFRI 1137269 | Nagarajan Nandagopal<br>Leah A Santat |
| Howard Hughes Medical Institute | Graduate Student Fellowship | Nagarajan Nandagopal |

The funders had no role in study design, data collection and interpretation, or the decision to submit the work for publication.

### Author contributions

Nagarajan Nandagopal, Leah A Santat, Conceptualization, Data curation, Formal analysis, Investigation, Methodology, Writing—original draft, Writing—review and editing; Michael B Elowitz, Conceptualization, Resources, Supervision, Funding acquisition, Investigation, Methodology, Writing—original draft, Project administration, Writing—review and editing

### Author ORCIDs

Nagarajan Nandagopal (iD) http://orcid.org/0000-0002-0469-6549
Leah A Santat (iD) http://orcid.org/0000-0003-0511-9740
Michael B Elowitz (iD) http://orcid.org/0000-0002-1221-0967

### Decision letter and Author response

Decision letter https://doi.org/10.7554/eLife.37880.023
Author response https://doi.org/10.7554/eLife.37880.024

## Additional files

### Supplementary files

• Transparent reporting form
DOI: https://doi.org/10.7554/eLife.37880.019

### Data availability

RNA sequencing data have been deposited in GEO under accession codes GSE113937. Source data files have been provided for Figure 5.

The following dataset was generated:

| Author(s) | Year | Dataset title | Dataset URL | Database and Identifier |
|---|---|---|---|---|
| Nandagopal N | 2018 | Gene expression in cultured mouse neural progenitor cells | https://www.ncbi.nlm.nih.gov/geo/query/acc.cgi?acc=GSE113937 | NCBI Gene Expression Omnibus, GSE113937 |

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
