## [Decision Letter]

Thank you for submitting your article "*Cis*-activation in the Notch signaling pathway" for consideration by *eLife*. Your article has been reviewed by two peer reviewers, and the evaluation has been overseen by Naama Barkai as the Senior and Reviewing Editor. The following individual involved in review of your submission has agreed to reveal his identity: Andre Levchenko (Reviewer #2).

The reviewers have discussed the reviews with one another and the Reviewing Editor has drafted this decision to help you prepare a revised submission.

The authors present evidence for a new mechanism of Notch receptor activation by which cis-ligands can activate Notch receptors on the same cell. They find that cis-activation is governed by ligand concentration, cis-inhibition, and Fringe glycosylation, and extends to Dll1/Dll4 and Notch1/2 receptor pairs. Notch ligands and receptors co-expressed in the same cell have been known to mutually inhibit one another, suppressing productive intercellular signaling the fact that cis-ligands can also activate Notch receptors is surprising and potentially highly significant. The authors further simulate scenarios to best describe the complexes and kinetics that can contribute to the co-existence of both cis- and trans-activation, and postulate on how this new mechanism may expand the capabilities of Notch signaling. This finding adds another level of complexity to the Notch signaling pathway that might have important consequences in other biological contexts.

Although many aspects of the underlying mechanisms remain puzzling, this manuscript presents a body of very well done work, including data not only characterizing the phenomenon, but also suggesting its functional significance (e.g., enhancing survival of neural stem cells). I do have several comments, focusing on points requiring further analysis and clarification:

The major limitation of this work is that the majority of observations were derived from synthetic Notch receptors in CHO cells. It is therefore of critical importance that the authors rigorously demonstrate that such a mechanism is utilized naturally. Specifically, please address the following essential comments:

1) In Figure 1—figure supplement 3, CRISPR deletion of *Notch2* and Jagged1 are performed in murine NMuMG cells, but the relevant manipulations are made using synthetic Dll1 and Notch1 constructs. The reasoning for this is unclear. Western blot profiling of NMuMG lysates showing expressed Notch receptors and ligands at the protein level, along with successful CRISPR depletion is needed. Genetic depletion followed by expression of the corresponding wild type receptor and inducible wild type ligand should be performed. Does eventual cis-inhibition occur (Figure 1—figure supplement 3F)? And if so, at what expression level in comparison to endogenously expressed ligand?

2) The NSC experiments in Figure 3 rely entirely on the γ-secretase inhibitor DAPT. Γ-secretase has numerous substrates outside of the Notch receptors and their ligands that can contribute to decreased proliferation and differentiation, and does not directly link this effect to Notch cis-activation.

3) Figure 1—figure supplement 4, the Caco-2 experiments rely on DAPT (100 uM!). Again, this indirect inhibitor does not directly implicate Notch activation. Genetic depletion followed by expression of wild type receptor and inducible wild type ligand would be ideal (as in Figure 1—figure supplement 3A).

In addition, please consider the following suggestions:

1) It is interesting that all cis-activation assays were completed in monolayer cultures using a small percentage of expressing cells. It would seem that manipulating ligand expression in single, isolated cells would most rigorously demonstrate this mechanism, albeit at lower throughput. Is there some element of cell-cell contact that is required? It is a bit concerning that the synthetic Notch receptors show both impaired and cell-contact dependence in their localization (Figure 1figure supplement 3D) compared to a more complete wild type Notch receptor, particularly as the authors describe that cis-activation occurs on the cell membrane and not in endosomal compartments (Figure 4). This further illustrates the need to clearly demonstrate this mechanism with wild type receptors, ligands, and in human cells.

2) Subsection “*Notch2* shows stronger cis-activation but decreased cis-inhibition compared to Notch1”: In Figure2D, the profile of activation was monotonic, and it was stated that Notch 2 is not cis-inhibited as strongly as Notch 1. If the cis-inhibition needs the formation of larger ligand-receptor cluster as mentioned in Discussion, does it imply Notch 2 his less/more effective as a molecule nucleating such clusters?

3) Discussion, fourth paragraph: It is mentioned that autocrine and juxtacrine signaling modes lead to different biological outcomes. It would be great if authors briefly summarize what the outcomes are.

4) Subsection “Cis-activation regulates neural stem cell maintenance and differentiation”, last paragraph: It is claimed that DAPT treatment altered expression patterns of markers in Figure 3—figure supplement 2B. However, the patterns of untreated and DAPT look overlapping in the figure. Please specify the degree of the presumed alteration, and how one can decipher it.

5) Figure 1—figure supplement 2B: As the percentage of engineered cells in wild type cell population is critical in this study, it will be more intuitive to represent the cell number of Y-axis in the form percentage or ratio.

6) Subsection “Notch1-Dll1 cells show ligand-dependent cis-activation”, sixth paragraph: The way authors interpret Figure 1—figure supplement 4, it would appear that cis-activation in sparsely plated CaCo-2 is similar to cis +trans activation in densely plated CaCo-2. First, it is not obvious because Figure1—figure supplement 4A and B are represented as the fold change relative to DAPT control in each condition. Please represent the values normalized to the same control. Also, please mention it in the figure caption. Second, this result is different from the previous results in Figure1—figure supplement 2C which showed that cis-trans activation is much higher than cis activation. This needs further analysis/explanation.

[Editors' note: further revisions were requested prior to acceptance, as described below.]

Thank you for resubmitting your work entitled "Cis-activation in the Notch signaling pathway" for further consideration at *eLife*. Your revised article has been favorably evaluated by Naama Barkai (Senior Editor) and two reviewers.

The reviewers would like to see your paper published in *eLife*, conditioned that you add a western blot that shows the extent of Dll1 depletion in Figure 3 and supports the claim that "Dll1 siRNA treatment also decreased survival, albeit more weakly, likely due to the incomplete knockdown (Figure 3C, right panel)".

*Reviewer #1:*

The manuscript has been improved but there are some remaining issues that need to be addressed before acceptance, as outlined below.

Major concerns regarding the use of synthetic receptors and ectopic ligands and their proper localization are adequately addressed within the new manuscript. The authors have expanded their original document to include Notch activation in sparse cells with endogenous receptors and ligands as well as ectopic ligands, as in Figure 1—figure supplement 4. Importantly, the authors also provide insight into what expression levels relative to endogenous Dll1 contribute to cis-activation and -inhibition, respectively.

The concerns regarding the use of DAPT as the sole method to explore the effect of Notch cis-activation on proliferation has been satisfactorily addressed in the document with experiments using silencing RNA to Dll1, Dll1 surface plating, and lower concentrations of DAPT (10uM, Figure 1—figure supplement 4). While, we find this approach satisfactory, a Western blot is necessary to show extent of Dll1 depletion in Figure 3 and to support the following claim "Dll1 siRNA treatment also decreased survival, albeit more weakly, likely due to the incomplete knockdown (Figure 3C, right panel)".

*Reviewer #2:*

I reviewed the manuscript and the answers. Basically, the authors revised the manuscript according to only two of my initial comments. I think those parts were revised properly. But, for the other three comments, they just retracted their claims, removed the figures, or stated that "they would like to follow up on it in the future". Although it might not be the best way to handle the criticisms, I am okay with these changes, considering that they do not undermine their major claims or change the overall message. Overall, it is a well thought-through story and the edits tightened it up and focused on the main claims, which are well supported. I am ok with giving the manuscript the green light for acceptance.

---

## [Author Response]

[…] Although many aspects of the underlying mechanisms remain puzzling, this manuscript presents a body of very well done work, including data not only characterizing the phenomenon, but also suggesting its functional significance (e.g., enhancing survival of neural stem cells). I do have several comments, focusing on points requiring further analysis and clarification:The major limitation of this work is that the majority of observations were derived from synthetic Notch receptors in CHO cells. It is therefore of critical importance that the authors rigorously demonstrate that such a mechanism is utilized naturally. Specifically, please address the following essential comments:1) In Figure 1—figure supplement 3, CRISPR deletion of Notch2 and Jagged1 are performed in murine NMuMG cells, but the relevant manipulations are made using synthetic Dll1 and Notch1 constructs. The reasoning for this is unclear.

We agree this was unclear as originally presented. The goal of the experiment was to determine whether engineered Notch1 and Dll1 constructs, which were shown to cis-activate in CHO-K1 could also cis-activate in a different cell type (NMuMG cells). Deletion of *Notch2* and Jagged1 was done to eliminate the potential for competition between the ectopic components (Dll1, Notch1) and the endogenous components (principally Jag1, *Notch2*). We now show (Figure 1—figure supplement 3B, right panel) that deletion of these components is required to observe signaling between the ectopic Dll1 and Notch1. To address this issue, we clarified the reasoning behind these experiments in the text. Furthermore, we also show that endogenous *Notch2* and Jag1 can cis-activate (see below).

Western blot profiling of NMuMG lysates showing expressed Notch receptors and ligands at the protein level, along with successful CRISPR depletion is needed.

We now show, through Western Blot analysis, the successful CRISPR deletion of *Notch2* and Jagged1 protein in our NMuMG cells (Figure 1—figure supplement 3B, middle).

Genetic depletion followed by expression of the corresponding wild type receptor and inducible wild type ligand should be performed.

We make three points in the NMuMG context:

1) Cis-activation between ectopic ligand and receptor.By deleting endogenous Jag1 and *Notch2*, and ectopically expressing Dll1 and Notch1, we show that cis-activation can occur between the ectopic ligand and receptor in a cell type distinct from CHO-K1 (Figure 1—figure supplement 3F, G). This result was included in the original version.

2) Cis-activation between endogenous receptor and ectopic ligand. Endogenous *Notch2* can be *cis-*activated by ectopic expression of Dll1, as shown using RT-qPCR and single-cell HCR-FISH measurements of Hes1 with or without DAPT treatment (Figure 1—figure supplement 4B-D). This result is new to this version.

*3)* Cis-activation between endogenous receptor and endogenous ligand. Consistent with *cis-*activation between endogenous Jag1 and *Notch2*, Hes1 expression also exhibits a modest reduction in response to DAPT addition in the absence of any ectopic components (Figure 1—figure supplement 4B-D). This result is new to this version.

The above mentioned results, as explained in the manuscript, support the claim that *cis*-activation occurs endogenously as well as with ectopic components in isolated NMuMG cells.

Does eventual cis-inhibition occur (Figure 1—figure supplement 3F)? And if so, at what expression level in comparison to endogenously expressed ligand?

We addressed this question with two new experiments:

First, to address this question, we extended the range of Dll1 expression levels by transiently transfecting a Dll1-mCherry construct and gating the flow cytometry data on different mCherry windows. These experiments revealed a non-monotonic dependence on Dll1 expression (Figure 1—figure supplement 3G), similar to that observed in CHO-K1 cells.

Second, using RNAseq, we compared the expression level of ectopic Dll1 in NMuMG N1D1+Rfng cells to endogenous Jag1 expression in wild-type NMuMG cells. As shown in Figure 1—figure supplement 3G (black bars on right), the levels of ectopic Dll1 expression that show peak cis-activation responses are similar to the expression levels of endogenous Jag1. Above this level, one starts to observe cis-inhibition (decreasing green bars). These results suggest that the levels of Dll1 required for cis-activation are comparable to endogenous ligand levels, and also support cis-inhibition dominating at higher ligand expression levels.

2) The NSC experiments in Figure 3 rely entirely on the γ-secretase inhibitor DAPT. Γ-secretase has numerous substrates outside of the Notch receptors and their ligands that can contribute to decreased proliferation and differentiation, and does not directly link this effect to Notch cis-activation.

In NSCs, we sought to test whether increasing or decreasing Notch signaling could lead to increased or decreased proliferation, respectively. Previously, we used DAPT to decrease signaling and observed a corresponding decrease in cell proliferation. We also demonstrated that NSCs plated on recombinant Dll1ext-IgG leads to a striking increase in cell numbers. Thus modulation of Notch signaling in either direction produced a corresponding effect on proliferation.

To address the reviewer's concern about potential off target effects of DAPT, we added new experiments using siRNA to knock down Dll1 expression. Dll1 knockdown decreased survival of isolated NSCs in low growth factor conditions, consistent with expectations from the DAPT results (Figure 3C). The magnitude of this effect is weaker than obtained with DAPT, likely because of incomplete knockdown. Nevertheless, together with the previous results, these findings are consistent with Notch signaling in isolated cells (cis-activation) impacting proliferation (Figure 3C and D). They are also consistent with previous work showing a positive effect of Notch signaling on proliferation in NSCs, particularly in the absence of growth factors (Nagao et al., 2007). In addition to including the new results in the manuscript, we also now explicitly emphasize that while we cannot rule out a non-specific effect of DAPT, our results are consistent with an effect of Notch on proliferation.

3) Figure 1—figure supplement 4, the Caco-2 experiments rely on DAPT (100 uM!). Again, this indirect inhibitor does not directly implicate Notch activation. Genetic depletion followed by expression of wild type receptor and inducible wild type ligand would be ideal (as in Figure 1—figure supplement 3A).

We originally chose to use a DAPT concentration of 100 µM based on previous work analyzing Notch signaling in Caco-2 cells (Hsieh, En Hui and Lo, 2012). However, we have re-performed the Caco-2 experiments using 10µM DAPT (similar DAPT levels were used for the NMuMG and NSC experiments) and observed similar results. The corresponding figure has now been updated to reflect the new data (Figure 1—figure supplement 4A). We agree that the ideal experiment would be done with specific genetic manipulations. However, Caco-2 cells are very slow growing (doubling time is 3-4 days), and therefore genetic manipulation would require extended timescales before resubmission.

In addition, please consider the following suggestions:1) It is interesting that all cis-activation assays were completed in monolayer cultures using a small percentage of expressing cells. It would seem that manipulating ligand expression in single, isolated cells would most rigorously demonstrate this mechanism, albeit at lower throughput. Is there some element of cell-cell contact that is required?

We agree that analysis of isolated cells is most compelling. Originally, the surrounding CHO-K1 cells were used to reduce cell movement. To confirm that these cells are not necessary, we now show that the same *cis*-activation behavior occurs in N1D1+Rfng cells plated sparsely, without surrounding wild-type CHO-K1 cells. These results indicate that the phenomenon does not depend on the overall cell density (Figure 1—figure supplement 2B).

Further, the two results together strengthen the conclusions of the paper: Mammalian cells are rarely in isolation in natural contexts. The appearance of cis-activation in densely cultured cells shows that it is not an artifact of isolation.

It is a bit concerning that the synthetic Notch receptors show both impaired and cell-contact dependence in their localization (Figure 1figure supplement 3D) compared to a more complete wild type Notch receptor, particularly as the authors describe that cis-activation occurs on the cell membrane and not in endosomal compartments (Figure 4). This further illustrates the need to clearly demonstrate this mechanism with wild type receptors, ligands, and in human cells.

The points about subcellular localization were confusing as originally presented, and we have now clarified these points in the text. Briefly:

- In CHO-K1 cells, we observed no discernable difference in localization between wild-type and Notch1ECD-Gal4 receptors (Figure 1—figure supplement 3E), consistent with the non-polarized nature of CHO-K1 cells.

- However, in NMuMG cells, which do polarize, we found that the ankyrin domain present in the wild-type Notch ICD was necessary for proper localization, as expected based on previous work (Sasaki et al., 2007; Benhra et al., 2010 and 2011). Including this in the Gal4 receptor restored localization and function.

Thus, the localization issue with the synthetic NotchGal4 system was specific to NMuMG and could be circumvented by including more of the WT receptor intracellular domain. We now clarify this point in the text (subsection “Notch1-Dll1 cells show ligand-dependent cis-activation”, fifth paragraph).

2) Subsection “Notch2 shows stronger cis-activation but decreased cis-inhibition compared to Notch1”: In Figure2D, the profile of activation was monotonic, and it was stated that Notch 2 is not cis-inhibited as strongly as Notch 1. If the cis-inhibition needs the formation of larger ligand-receptor cluster as mentioned in Discussion, does it imply Notch 2 his less/more effective as a molecule nucleating such clusters?

This suggestion that the difference in signaling may reflect differences in cluster nucleation is very interesting. While we were not able to address it here, we would like to follow up on it in the future with better assays for cluster formation.

3) Discussion, fourth paragraph: It is mentioned that autocrine and juxtacrine signaling modes lead to different biological outcomes. It would be great if authors briefly summarize what the outcomes are.

We have added a brief summary of the differing biological outcomes that autocrine and juxtacrine signaling modes can lead to. Briefly, HB-EGF can occur as two isoforms: a membrane-anchored precursor that is involved in juxtacrine signaling or a cleaved soluble form that is involved in autocrine signaling. In MDCK cells, distinct cell phenotypes were found to occur depending upon which HB-EGF isoform was expressed, and cell survival and proliferation appeared to increase with the presence of the membrane-anchored isoform (Raab and Klagsbrun 1997; Singh, Sugimoto, and Harris 2007). Similarly, in yeast, rewiring of the mating pathway to create an autocrine signaling system revealed that qualitatively different behaviors ranging from quorum sensing to bimodality could be generated by tuning the relative strengths of *cis* and *trans* signaling (Youk and Lim 2014). These points are now included in the text.

4) Subsection “Cis-activation regulates neural stem cell maintenance and differentiation”, last paragraph: It is claimed that DAPT treatment altered expression patterns of markers in Figure 3—figure supplement 2B. However, the patterns of untreated and DAPT look overlapping in the figure. Please specify the degree of the presumed alteration, and how one can decipher it.

The effects on marker expression are modest at this time-point. The difficulty here is that larger effects on differentiation require longer durations, by which point cell divisions occur and we can no longer eliminate possible intercellular signaling. Based on these considerations, we removed this figure and the corresponding claims about differentiation.

5) Figure 1—figure supplement 2B: As the percentage of engineered cells in wild type cell population is critical in this study, it will be more intuitive to represent the cell number of Y-axis in the form percentage or ratio.

We agree and appreciate the suggestion. The changes are incorporated in the figures.

6) Subsection “Notch1-Dll1 cells show ligand-dependent cis-activation”, sixth paragraph: The way authors interpret Figure 1—figure supplement 4, it would appear that cis-activation in sparsely plated CaCo-2 is similar to cis +trans activation in densely plated CaCo-2. First, it is not obvious because Figure1—figure supplement 4A and B are represented as the fold change relative to DAPT control in each condition. Please represent the values normalized to the same control. Also, please mention it in the figure caption. Second, this result is different from the previous results in Figure1—figure supplement 2C which showed that cis-trans activation is much higher than cis activation. This needs further analysis/explanation.

Because Caco-2 cells grow slowly (as mentioned above), both low and high density conditions were in fact much lower than corresponding densities in the CHO-K1 experiments. In fact, the higher density condition in the Caco-2 experiments is still quite sparse and therefore behaves similarly to the lower density condition. For this reason, and because the density dependence in Caco-2 was not essential for main claims, we removed it in the revised version.

[Editors' note: further revisions were requested prior to acceptance, as described below.]

The reviewers would like to see your paper published in eLife, conditioned that you add a western blot that shows the extent of Dll1 depletion in Figure 3 and supports the claim that "Dll1 siRNA treatment also decreased survival, albeit more weakly, likely due to the incomplete knockdown (Figure 3C, right panel)".

Reviewer #1:

The manuscript has been improved but there are some remaining issues that need to be addressed before acceptance, as outlined below.Major concerns regarding the use of synthetic receptors and ectopic ligands and their proper localization are adequately addressed within the new manuscript. The authors have expanded their original document to include Notch activation in sparse cells with endogenous receptors and ligands as well as ectopic ligands, as in Figure 1—figure supplement 4. Importantly, the authors also provide insight into what expression levels relative to endogenous Dll1 contribute to cis-activation and -inhibition, respectively.The concerns regarding the use of DAPT as the sole method to explore the effect of Notch cis-activation on proliferation has been satisfactorily addressed in the document with experiments using silencing RNA to Dll1, Dll1 surface plating, and lower concentrations of DAPT (10uM, Figure 1—figure supplement 4). While, we find this approach satisfactory, a Western blot is necessary to show extent of Dll1 depletion in Figure 3 and to support the following claim "Dll1 siRNA treatment also decreased survival, albeit more weakly, likely due to the incomplete knockdown (Figure 3C, right panel)".

We thank you and the reviewers for the favorable review of the manuscript. As requested by reviewer 1, we have now updated Figure 3 to include a western blot that shows the level of Dll1 after siRNA treatment in neural stem cells. Specifically, this data shows that Dll1 protein is reduced after siRNA treatment, but not completely eliminated, supporting the claim "Dll1 siRNA treatment also decreased survival, albeit more weakly, likely due to the incomplete knockdown of Dll1 protein levels". We have now updated the relevant text to include a citation to the new data, and updated the corresponding figure legend and Materials and methods section to reflect the addition.